# Three-dimensional scene boundary representations for wall orientation and distance are represented distinctly in the human visual cortex

Yichen Wu[1,2,3,4], Sheng Li [iD][1,2,3,4]*

**1** School of Psychological and Cognitive Sciences, Peking University, Beijing, China, **2** PKU-IDG/ McGovern Institute for Brain Research, Peking University, Beijing, China, **3** Beijing Key Laboratory of Behavior and Mental Health, Peking University, Beijing, China, **4** Key Laboratory of Machine Perception (Ministry of Education), Peking University, Beijing, China

* sli@pku.edu.cn

## Abstract

Human spatial navigation relies on the brain's ability to visually represent the 3D layout of the environment. To understand how the brain encodes the layout information, it is crucial to identify the key features of environmental layout and how they are processed in the human brain. The vector coding principle, which highlight the role of boundary distance and orientation, provides a theoretical framework supported by physiological evidence from rodents. In this study, we developed a reconstruction approach to quantitatively estimate 3D layout information from natural indoor scene images. This approach enabled analyses of fMRI data from the large-scale Natural Scenes Dataset (NSD) using vector-based models of 3D layout. To validate the NSD-based results and examine task-related dynamics, we further conducted fMRI and MEG experiments with navigation-related and non-navigational tasks. Controlling for low-, mid-, and high-level visual and semantic features of natural indoor scenes, we found a spatiotemporal dissociation between boundary distance and orientation representations in the human brain. Relative distance was encoded in the early visual cortex during early processing in a task-invariant manner, whereas orientation was represented in scene-selective higher visual areas during later processing and was modulated by navigation-related tasks. Importantly, task modulation manifested as enhanced orientation coding in early visual cortex, potentially reflecting top-down feedback and short-term maintenance mechanisms. Together, these findings provide a novel perspective on how the human brain represents navigation-relevant information about the immediate surrounding environment, advancing our understanding of the neural mechanisms that link perception to action in spatial navigation.

**Data availability statement:** Data and script can be found at OSF at https://osf.io/uxwr4/.

**Funding:** This study was supported by grants from STI2030-Major Projects (2021ZD0200204, https://en.most.gov.cn/) and the National Natural Science Foundation of China (32271104, https://www.nsfc.gov.cn/english/site_1/index.html) to SL. The funders did not play any role in the study design, data collection and analysis, decision to publish, or preparation of the manuscript.

**Competing interests:** The authors have declared that no competing interests exist.

**Abbreviations:** CBOW, continuous bag of words; COCO, Common Objects in Context; EPI, echo-planar imaging; FDR, false discovery rate; fMRI, functional Magnetic Resonance Imaging; fROIs, functional regions of interest; GLM, General Linear Model; HRF, Hemodynamic Response Function; ICA, independent component analysis; MEG, magnetoencephalography; NSD, Natural Scenes Dataset; OPA, occipital place area; PPA, parahippocampal place area; RDMs, representational dissimilarity matrices; ROI, region of interest; RSA, representational similarity analysis; RSC, retrosplenial complex; SVR, support vector regression; TE, echo time; TFCE, threshold-free cluster enhancement; TR, repetition time.

## Introduction

Spatial navigation depends on the brain's ability to explore and represent the external world. Seminal discoveries in rodents, particularly the identification of place cells [1] and grid cells [2], have laid the foundation for understanding the neural mechanisms underlying spatial map construction. In humans, navigation relies predominantly on vision to represent the complex three-dimensional (3D) world, especially the spatial geometry of surrounding environment [3]. This visual representation of environmental layout is essential for reconstructing egocentric boundaries, which in turn form the basis for constructing allocentric cognitive maps [4]. In the human brain, a network of scene-selective areas, including the occipital place area (OPA), parahippocampal place area (PPA), and retrosplenial complex (RSC), has been identified as crucial for representing the local environment. These regions respond more strongly to images of landmarks, buildings, and rooms than to faces or objects, underscoring their central role in processing structural layout information [5].

Boundaries define the limits of exploration within an environment, making them critical elements of spatial representation in the brain. In a given space, walls serve as invariant components that determine the overall layout, whereas most objects are movable and subject to change. Thus, the arrangement of walls in a scene can be regarded as a primary representation of spatial layout. Previous studies on human spatial cognition have shown that scene-selective areas, particularly the OPA, are activated by walls [6–8] and encode navigationally relevant information [9–14]. More recent work using artificial stimuli to model 3D geometries further demonstrated that these areas also encode scene layout [15,16]. Notably, Lescroart and Gallant [16] found that scene-selective areas are tuned to the 3D configuration of surfaces in artificial scenes, even after controlling for low-level 2D visual features. This surface-based coding model provides valuable insights into how the human brain visually represents navigational information. However, surface-based models do not capture boundary vectors, a critical component of spatial representation identified in rodent navigation systems [17].

Successful navigation requires local representation of boundary information in an egocentric reference frame to construct the layout of agent's surroundings (e.g., front, back, left, and right). According to the principle of vector coding, neural representations of environmental features are organized by the direction and distance of boundaries relative to the agent, which together constitute the fundamental structure of spatial layout [17]. In rodents, neurons with vector-based receptive fields have been identified for boundary [18,19] and border [20–22], underscoring the essential roles of boundary vectors representations in spatial navigation. Despite these findings, how such vector-based coding is implemented in the human brain remains largely unknown.

Given the complexity with which the human brain transforms 2D natural images into 3D layout representations, large-scale neuroimaging datasets are particularly valuable for investigating these representations under naturalistic conditions. A growing trend in functional Magnetic Resonance Imaging (fMRI) research involves intensive scanning, which enables detailed analyses of large datasets from individual

participants [23]. A prominent example is the Natural Scenes Dataset (NSD), which adopts this approach with a recognition task using natural images [24]. The NSD provides a unique opportunity to study how the human brain represents spatial layout. However, unlike artificial stimuli, the natural images in the NSD lack ground-truth 3D layout annotations, limiting their use in fine-grained analyses of spatial cognition.

To address this limitation, we developed a computer vision-based approach to reconstruct 3D scene layouts from natural indoor images. Applying this method to the NSD enabled us to examine fine-grained neural representations of spatial layout in this large-scale 7T fMRI dataset. Because the NSD involved a non-navigational task, we further conducted two complementary experiments, using fMRI and magnetoencephalography (MEG), to validate and extend the NSD-based findings. These experiments incorporated both a navigation-related layout discrimination task and a non-navigational texture discrimination task, using naturalistic indoor images with ground-truth maps of boundary distance and orientation derived from the Matterport3D database. Across these datasets, we found converging evidence for a spatiotemporal dissociation between distance and orientation representations of layout boundaries, as well as task-dependent modulation in navigation-related context.

## Results

### A new approach for reconstructing 3D layout and self-pose from natural indoor images

Investigating how the brain represents 3D layout in large-scale natural image datasets (e.g., NSD) has been challenging because explicit layout annotations are typically available only for artificial stimuli. To overcome this limitation, we developed a reconstruction approach that quantitatively estimates both the 3D layout and self-pose of natural indoor images through an annotation-based procedure (Fig 1A, see Methods for details). Our approach uses a vanishing point-based camera calibration technique, leveraging the fact that parallel lines in the 3D world converge within the 2D projection of a scene. Six human annotators manually identified six key edges in each image. Assuming that the optical point coincides with the image center, we recovered essential camera parameters, including focal length, field of view, pitch, roll, and wall orientations. Based on these parameters, we generated 3D layout segmentation maps for 2,120 NSD indoor images (see Fig 1B for reconstruction examples).

To evaluated the robustness of our method, we tested on the Matterport3D database, which provides ground-truth camera parameters and layout annotations. Across 40 undistorted indoor images, the mean errors in pitch and roll angles were 1.65° (SD = 2.38°) and 1.13° (SD = 1.28°), respectively. Layout reconstruction accuracy, measured by the proportion of misclassified pixels relative to the ground-truth segmentation map, yielded a mean pixel error of 2.98% (SD = 3.64%). Because NSD stimuli were not distortion-corrected, the reconstruction accuracy is slightly reduced compared with the Matterport3D images, as the annotated lines that are supposed to be parallel to the ground may contain errors.

### Dissociated representations of boundary distance and orientation in the visual cortex: NSD experiment

The positions of side walls define the boundaries of an indoor space and are essential for ground-level navigation. From an egocentric aerial view, side walls can be modeled as orthogonal lines radiating from the observer (Fig 1B), a geometric structure that lies at the core of vector coding principle [17]. While our reconstruction approach estimates the orientations of side walls based on their intersection with the horizon, it does not directly recover precise distance. We therefore approximated the relative distance of side walls by computing the area difference between side wall regions and ceiling/floor regions. Specifically, these parameters reflect the comparative proximity of the left and right walls to the observer within the same indoor space. Under the rules of perspective, a larger image area occupied by a side wall indicates that the wall is relatively closer to the observer compared to the opposite wall. This relative distance information is crucial for self-localization, as it enables the observers to infer their specific position within the room with respect to its boundaries. The area difference parameters, together with reconstructed side wall orientations, were used to construct the layout models for walls' relative distance and orientation (Fig 2B).

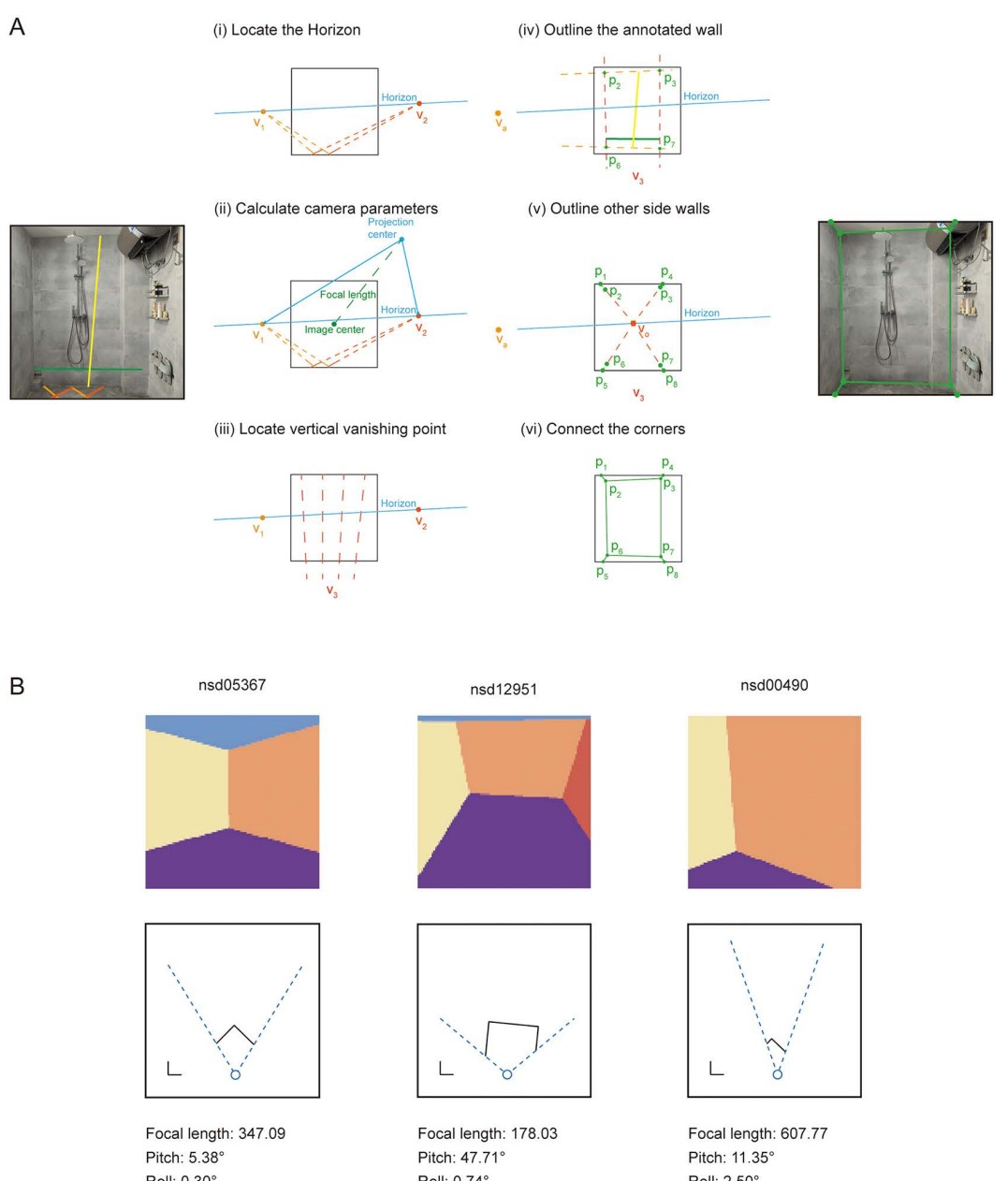

**Fig 1. Reconstruction of 3D layout and self-pose for natural indoor images. A**, Schematic of the reconstruction approach for an example image (see Methods for details). The left image shows the six manually labeled edges: the two pairs of orange line are the horizontal edges parallel to the ground, while the green line and yellow line were used to calculate the height and width of the middle wall, respectively. The middle panel illustrates the reconstruction procedure based on the labeled lines. The right image shows the reconstructed corners and wall conjunctions. The photograph was taken by the first author and is used for illustrative purposes only. **B**, Examples of reconstruction results. The top row displays the reconstructed layout segmentation maps, with their corresponding NSD indoor image IDs labeled above each map. The middle row presents the aerial-view maps with side walls marked by black lines. The blue circle indicates the observer, and the dashed blue lines define the field of view. Reconstructed parameters are shown at the bottom.

Layout features inevitably covary with 2D image statistics. We aimed to ensure that the hypothesized layout representations could not be fully explained by generic image features, a common concern in studies of visual navigation [9,15,16]. To this end, we incorporated three additional models to control for low-, mid-, and high-level visual and semantic features (Fig 2A): the GIST model for low-level features of spatial frequency and orientation [25], the S-P model for mid-level features of texture [26], and the object2vec model for high-level features of semantic and object cooccurrence [27]. We constructed representational dissimilarity matrices (RDMs) for each model and region of interest (ROI). We then conducted representational similarity analysis (RSA) using partial Spearman correlation to assess the unique contribution of each model while controlling for the others. By partitioning shared variance among competing models, this approach allowed us to quantify the variance in neural activity uniquely attributable to layout features, thereby mitigating potential confounds arising from non-navigational 2D image statistics.

We found that 2D visual and semantic features are significantly partial correlated with neural activities across both early visual and scene-selective areas (Fig 2B). The presence of semantic-related variance in early visual regions is consistent with recent findings that semantic dimensions can explain neural responses in these areas when naturalistic stimuli are used [28,29]. However, these effects should be interpreted with caution, as our analyses were based on a restricted subset of indoor scenes from the broader NSD dataset, sampling only a limited portion of the semantic space. Importantly, the overall correspondence between the GIST, texture, and semantic models and low-, mid-, and high-level visual regions confirmed that these models capture the intended representational hierarchy. This validation pattern indicates that the control models captured the intended visual and semantic structure of the stimuli, providing confidence that a substantial portion of non-layout variance was successfully accounted for, thereby enabling isolation of the unique contribution of 3D layout representations.

Critically, after controlling for these 2D visual and semantic features, early visual areas exhibited significant partial correlations with the relative distance model (V1: $t(7) = 9.72$, $q < 0.001$; V2: $t(7) = 7.05$, $q < 0.001$; V3: $t(7) = 4.95$, $q = 0.002$; hV4: $t(7) = 3.94$, $q = 0.006$; one-tailed, FDR corrected), whereas hV4, OPA and PPA showed significant partial correlations with the orientation model (hV4: $t(7) = 2.50$, $q = 0.029$; OPA: $t(7) = 2.67$, $q = 0.023$; PPA: $t(7) = 2.20$, $q = 0.042$; Fig 2D). As a control, the full Spearman correlation results for the layout models, computed without controlling for other variables, are presented in Fig 2D.

These results revealed a clear dissociation along the visual hierarchy: the early visual cortex predominantly represents boundary distance, whereas scene-selective regions encode boundary orientation. To confirm this dissociation at the individual-subject level, we performed a searchlight analysis within the visual cortex of each participant. Fig 3 illustrates an example participant's flattened cortical surface maps showing the distributions of correlation for the relative distance and orientation models after controlling for 2D visual and semantic features (see S3 Fig for maps of other NSD participants). The surface map demonstrates a posterior-anterior gradient from distance to orientation representation: voxels in the posterior occipital cortex uniquely correlated with the relative distance model, while anterior occipital and temporal regions showed stronger correlations with the orientation model. Importantly, these effects could not be accounted for by alternative 3D layout models, such as mean depth model or the models used in Henriksson and colleagues [15] and Lescroart and Gallant [16]; both boundary relative distance and orientation models remained significant after controlling for these alternatives (S2 Fig). Together, these analyses indicate distinct and hierarchically organized neural representations of boundary distance and orientation along the human visual processing stream.

## Representation of self-pose in the visual cortex: NSD experiment

Self-pose parameters, such as pitch and roll angles, characterize the orientation of the observer relative to the surrounding environment, thereby defining the egocentric viewing geometry from which a scene is perceived. These parameters are fundamental aspects of spatial cognition and arise from the integration of multisensory signals, among which visual input provides a significant source of information [30,31]. However, direct neural evidence regarding how the human

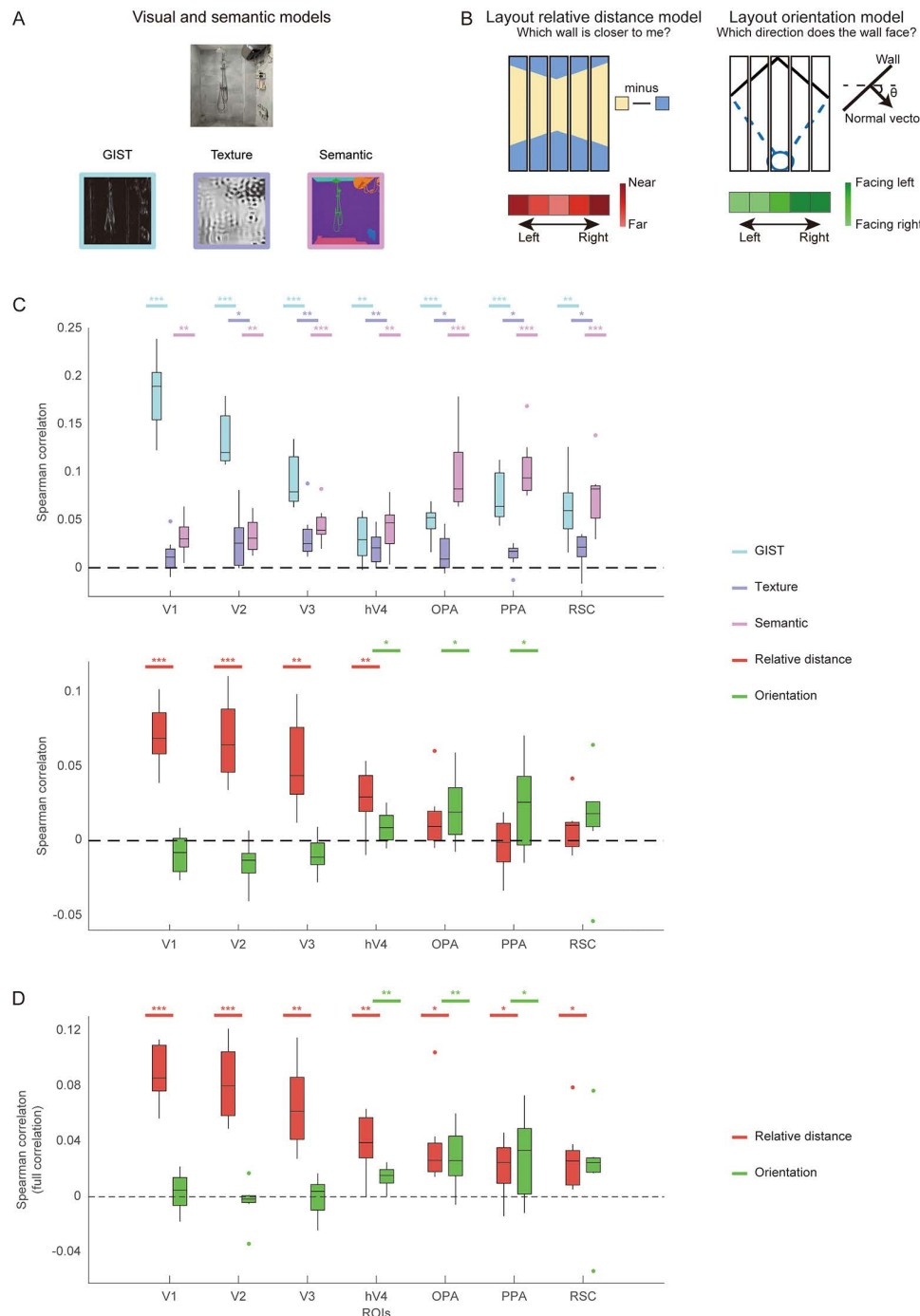

**Fig 2. Model representations for 2D features and 3D layout: NSD experiment. A**, Models for 2D visual and semantic features. **B**, Models for 3D layout features, including boundary relative distance and orientation. **C**, RSA results across ROIs. The upper plot shows the partial correlations between 2D model RDMs and neural RDMs. The lower plot shows the partial correlations between 3D model RDMs and neural RDMs. Both sets of results were derived from the same partial correlation analysis but are displayed separately for clarity. **D**, The full Spearman correlation RSA analysis without controlling image features. Asterisks indicate significance in one-tailed *t* test against chance level (0). * *q* < 0.05; ** *q* < 0.01, *** *q* < 0.001. The data underlying this figure can be found at https://doi.org/10.17605/OSF.IO/UXWR4.

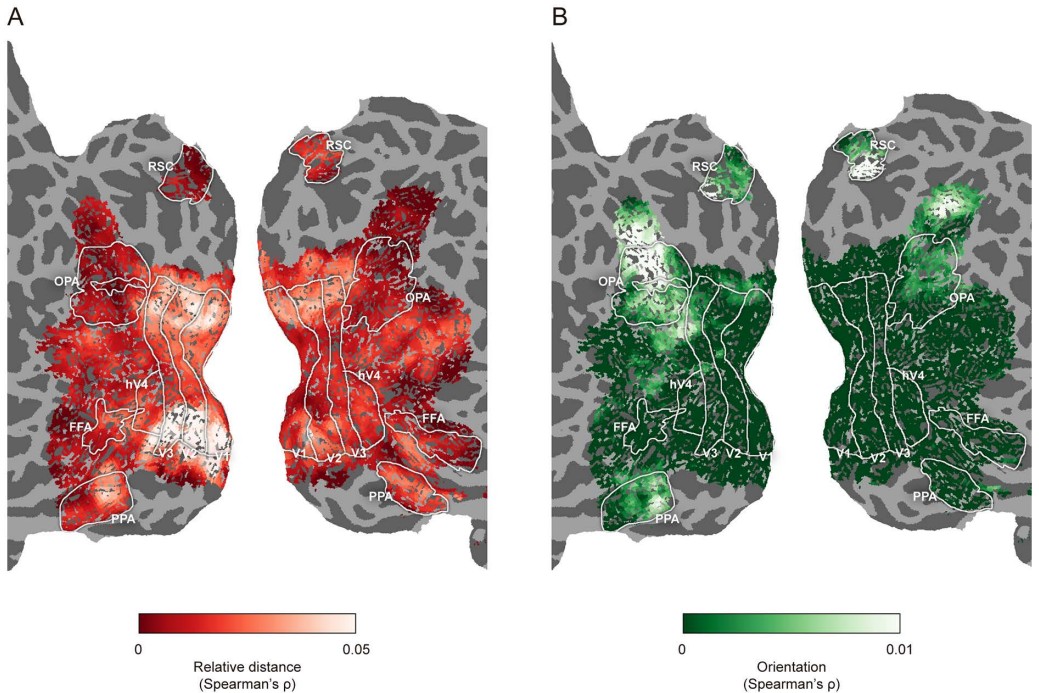

**Fig 3. Cortical representations of 3D layout for one example participant: NSD experiment (Subject 1 in NSD dataset).** Searchlight-based partial correlations are shown on flattened cortical surface maps (see S3 Fig for maps of other participants). **A**, Representation of the relative distance model. **B**, Representation of the orientation model.

visual cortex encodes self-pose information in naturalistic scenes remains limited. Using our scene-layout reconstruction approach, we estimated the self-pose parameters of NSD indoor images, with their distributions shown in Fig 4A. Using the reconstructed pitch and roll angles of NSD indoor images, we tested whether self-pose information could be decoded from the fMRI responses in early visual and scene-selective regions (Fig 4B).

Significant prediction of pitch angles was observed across all early visual and scene-selective regions (V1: $t(7)$ = 3.08, $q$ = 0.028; V2: $t(7)$ = 2.91, $q$ = 0.028; V3: $t(7)$ = 4.42, $q$ = 0.007; hV4: $t(7)$ = 2.87, $q$ = 0.028; OPA: $t(7)$ = 18.52, $q$ < 0.001; PPA: $t(7)$ = 18.52, $q$ < 0.001; RSC: $t(7)$ = 2.57, $q$ = 0.037). In contrast, roll angles could not be reliably decoded in any ROI, indicating that pitch, but not roll, is encoded in the visual cortex. This finding provides direct evidence that the visual system contains representations of navigation-relevant self-pose information, consistent with the behavioral importance of pitch during vertical navigation (e.g., ascending or descending stairs).

## Task-dependent enhancement of layout representation in early visual cortex: fMRI experiment

The NSD analyses revealed that both early visual and scene-selective areas encode layout and self-pose information even when scene layout is task-irrelevant. To determine how these representations are modulated when layout becomes behaviorally relevant, we conducted a new fMRI experiment. Participants performed two block-designed tasks: a layout discrimination task, which required active encoding of the spatial layout, and a texture discrimination task, which emphasized surface texture and color and served as a non-navigational control (Fig 5A).

Stimuli were drawn from the Matterport3D database [32], which provides ground-truth camera poses as well as depth, segmentation, and surface normal maps (Fig 5B). These annotations enabled precise construction of distance and orientation models for side walls. To avoid potential confounds from self-pose variation, all images were selected with pitch and

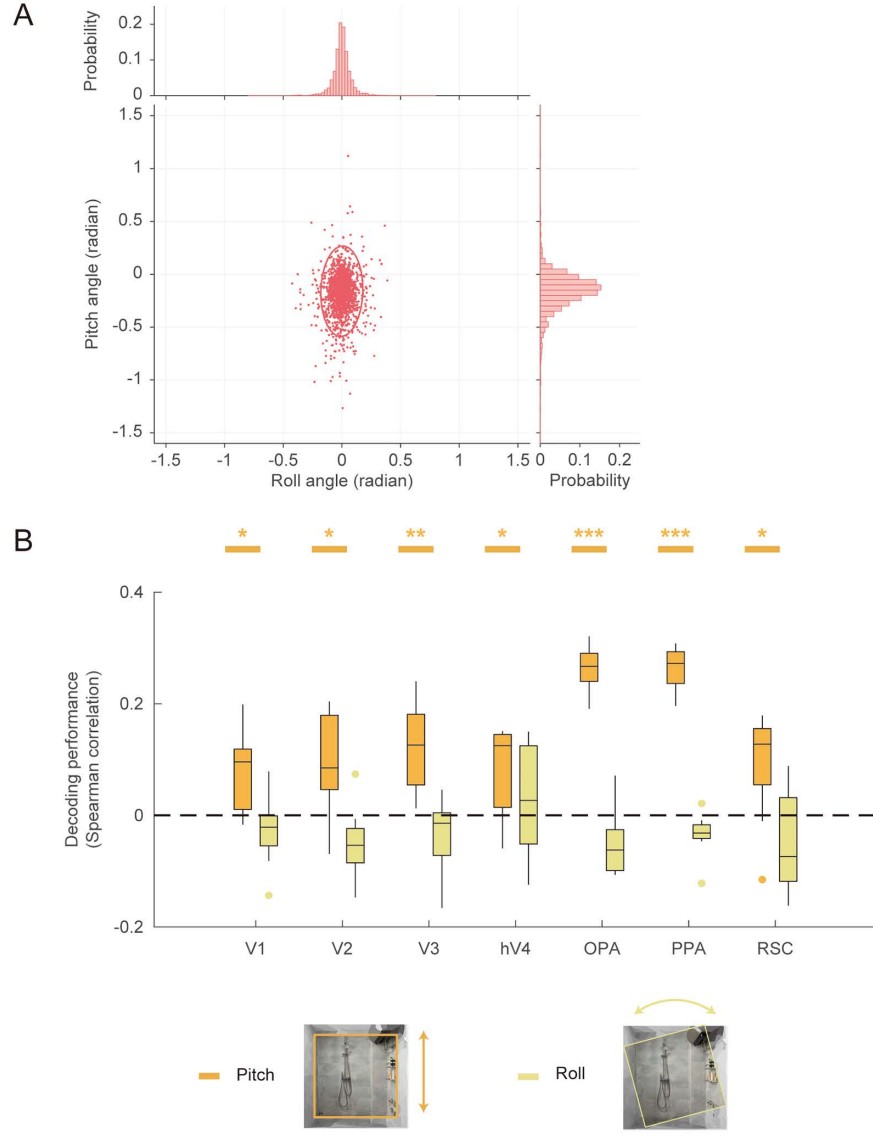

**Fig 4. Decoding self-pose from fMRI signals: NSD experiment. A**, Distributions of the pitch and roll angles of NSD indoor images. The circles in the scatterplot represent the 95% percentile of angles. **B**, Pitch and roll angles were decoded using support vector regression (SVR). Only the pitch angle showed reliable decoding across the visual cortex. Asterisks indicate significance in one-tailed $t$ test against chance level (0). * $q < 0.05$; ** $q < 0.01$; *** $q < 0.001$. The data underlying this figure can be found at https://doi.org/10.17605/OSF.IO/UXWR4.

roll angles fixed at 0°. Task difficulty was equated through a pilot experiment, with accuracies of 77.60% and 76.32% for layout and texture tasks, respectively ($t(29) = 0.41$, $p = 0.683$).

We performed RSA on pre-defined ROIs, including V1 and scene-selective areas (OPA, PPA, and RSC). Fig 6 shows partial correlations between neural RDMs and the layout models after controlling for 2D visual and semantic features. We observed robust task-dependent modulation of layout representations. In the layout discrimination task (Fig 6A, right), both V1 and OPA showed significant partial correlations with relative distance (V1: $t(29) = 8.41$, $q < 0.001$; OPA: $t(29) = 2.99$, $q = 0.011$) and orientation (V1: $t(29) = 5.61$, $q < 0.001$; OPA: $t(29) = 3.88$, $q = 0.001$) models. In contrast, in the texture discrimination task (Fig 6B, right), V1 and PPA showed significant partial correlations with the relative

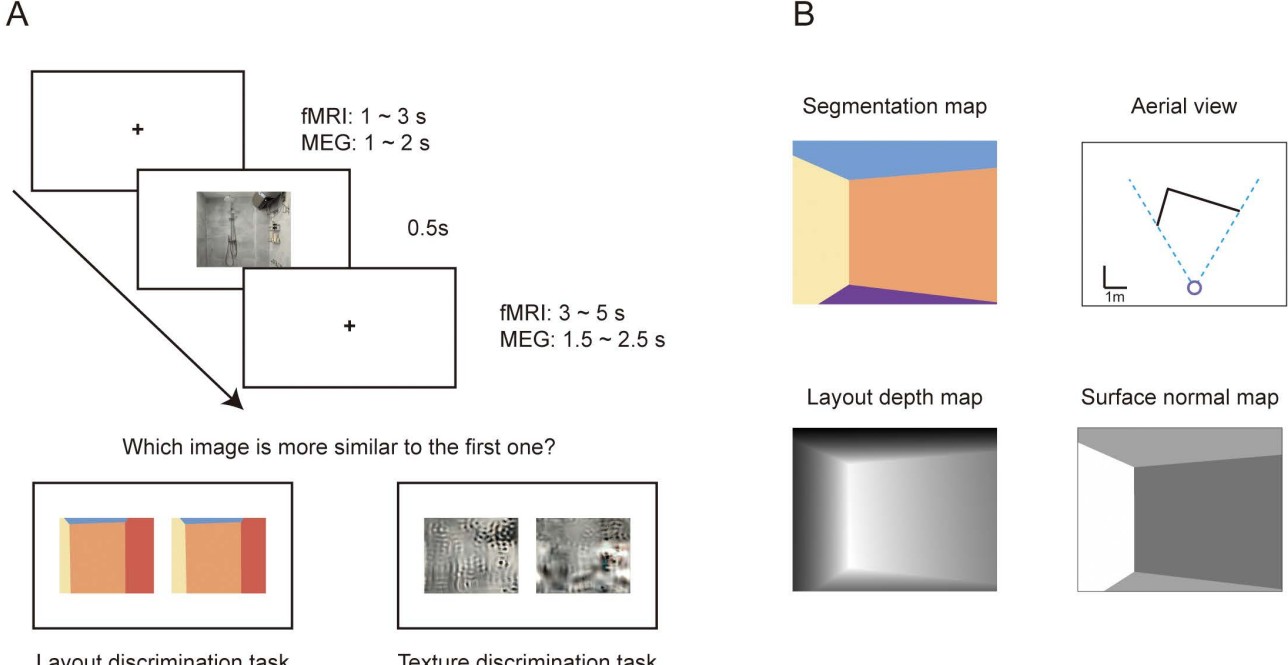

**Fig 5. Task procedures for the Matterport3D fMRI and MEG experiments. A**, Match-to-sample paradigms for layout discrimination and texture discrimination tasks. Inter-trial and inter-stimulus intervals are adjusted to meet the requirements of the fMRI and MEG. **B**, An example illustrating the ground-truth layout-related annotations for a scene image from the Matterport3D database.

distance model (V1: $t(29)$ = 4.68, $q<0.001$; PPA: $t(29)$ = 2.71, $q=0.016$), while only OPA showed a significant partial correlation with orientation ($t(29)$ = 2.78, $q=0.015$). Crucially, direct task comparison revealed significantly enhanced representation in V1 for both relative distance ($t(29)$ = 3.99, $q=0.002$, two-tailed, FDR corrected) and orientation ($t(29)$ = 5.21, $q<0.001$) models (Fig 6C).

To further examine whether the brain encodes metric depth information, we built a precise distance model using the ground-truth depth maps from Matterport3D. Unlike our area-based relative distance model, this metric-distance model represents the true physical distance from the observer in meters, independent of 2D image proportions. Consequently, two images that exhibit similar side-wall area in 2D image plane may nonetheless correspond to substantially different physical distances in 3D space, depending on camera parameters and the actual size of the room. Neither this precise distance model nor a mean depth model (S1 Text) showed significant correlation with neural activity (S4 Fig), suggesting that precise metric distance models may not be the optimal descriptors of wall distance representations.

Despite our comprehensive controls, it remains possible that the layout representations observed in V1 were influenced by visual or semantic factors not captured by the initial control models. One plausible factor is the local image contrast within the central visual field, a well-established driver of V1 activity. Another factor is the distribution of object clutter, which may covary with wall distance because deeper walls typically encompass a larger physical volume and thus allow more objects to be present. To address these possibilities, we conducted additional partial RSA analyses that explicitly controlled for central contrast and object clutter. As shown in S5 Fig, the neural representations of relative distance and orientation in V1 and scene-selective regions remained statistically significant after accounting for these factors. These results suggest that the observed layout encoding cannot be reduced to local contrast differences or object- density variations alone.

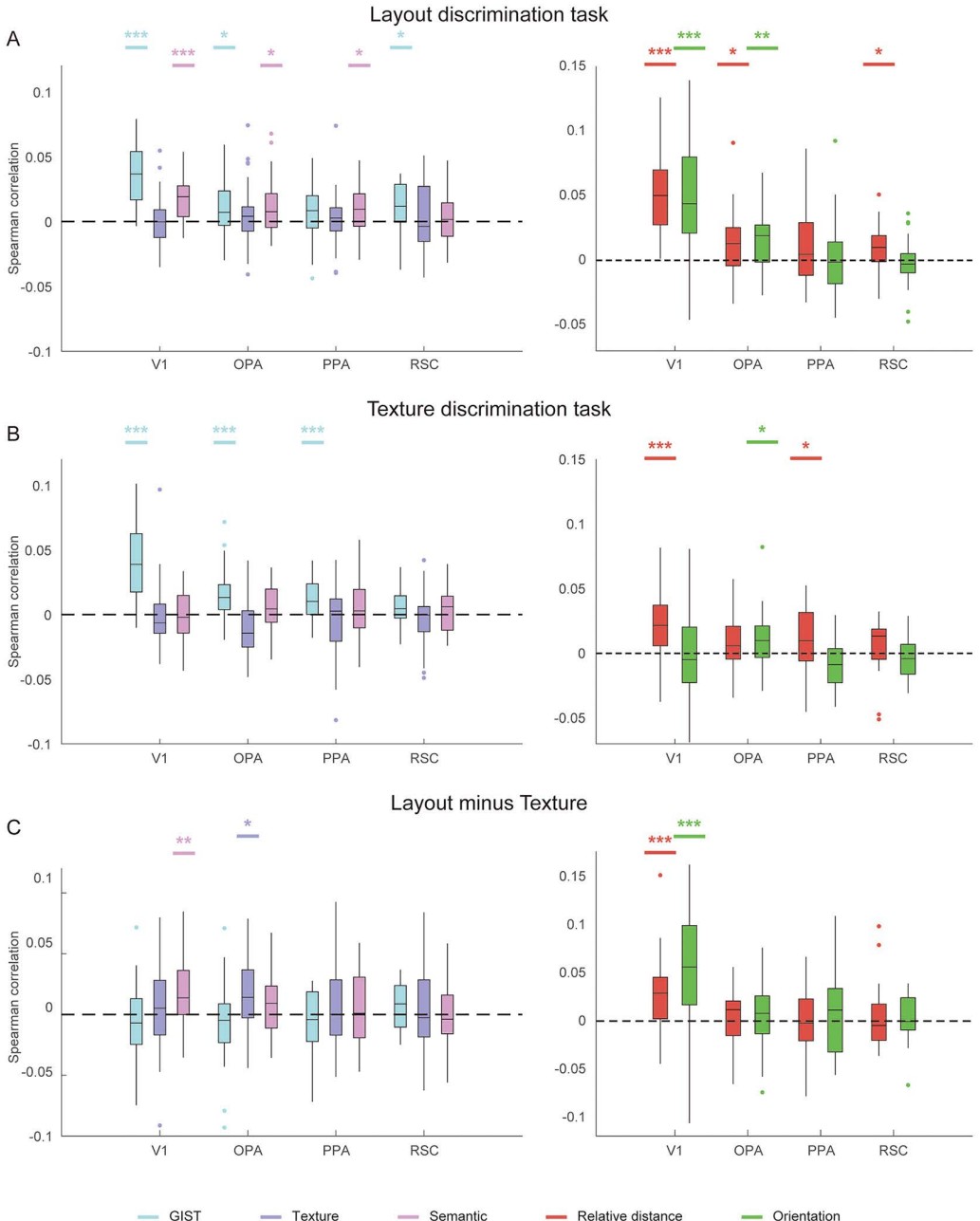

**Fig 6. Model representations for 2D features and 3D layout: Matterport3D fMRI experiment.** The left plots show partial correlations between 2D model RDMs and neural RDMs (see S1 Text for statistics). The right plots show partial correlations between 3D layout model RDMs and neural RDMs. **A**, Layout discrimination task. **B**, Texture discrimination task. **C**, Task-dependent enhancement of representation, calculated as the difference between partial correlation coefficients for the two tasks (layout minus texture). Asterisks in **A** and **B** denote significant results in one-tailed $t$ test against chance (0), while asterisks in **C** indicate significance based on two-tailed tests. * $q < 0.05$; ** $q < 0.01$, *** $q < 0.001$. The data underlying this figure can be found at https://doi.org/10.17605/OSF.IO/UXWR4.

Together, these results highlight two key implications. First, the texture discrimination task replicates the NSD-based findings, supporting task-invariant encoding of spatial layout in the visual cortex. Second, the layout discrimination task reveals enhanced representations of boundary distance and orientation when layout information becomes task-relevant, suggesting possible feedback modulation in V1. However, the limited temporal resolution of fMRI precludes direct characterization of such feedback processes, motivating a subsequent MEG experiment.

## Temporal dissociation of distance and orientation representations: MEG experiment

To characterize the temporal dynamics of layout representations and their task-dependent modulation, we conducted an MEG experiment using the same stimuli and design as in the fMRI experiment. Participants achieved comparable accuracies across tasks (layout: 72.73%; texture: 72.52%; $t(31) = 0.18$, $p = 0.856$).

For each task, we constructed a neural RDM at each time point of the MEG signal using occipital channels. RSAs were performed between neural RDMs and model RDMs. Partial correlation analysis revealed significant temporal clusters correlated with the GIST and semantic models in the layout discrimination task ($p < 0.05$, TFCE one-tailed; Fig 7A, left) and with the GIST model along in the texture discrimination task ($p < 0.05$; Fig 7B, left).

Critically, relative distance representations emerged in both tasks (Fig 7A and 7B, right): 160–240 ms and 270–380 ms for the layout discrimination task, and 140–230 ms, 550–620 ms, and 730–760 ms for the texture discrimination task. In contrast, orientation representations appeared exclusively in the layout discrimination task within later time windows (410–560 ms and 670–720 ms), demonstrating a temporal dissociation between distance and orientation coding. Eye-tracking data collected during the experiment confirmed that participants maintained stable central fixation. The average farthest fixation distance across all participants was 1.330° (standard deviation: 0.527°). Subsequent partial correlation RSA further demonstrated that the fixation patterns could not account for the layout representations observed in the MEG signals (S1 Text and S6 Fig).

To quantify task-dependent effects, we focused on three 100-ms windows centered at 200, 500, and 700 ms (Fig 7C), which corresponded to time periods showing significant RSA correlations with the layout models. Task-dependent enhancement was evident at 500 ms and 700 ms windows (450–550 ms: $t(31) = 2.85$, $q = 0.023$; 650–750 ms: $t(31) = 3.06$, $q = 0.023$; two-tailed, FDR corrected). Searchlight analysis across channels corroborated these results (Fig 7D): early clusters (~200 ms) over posterior channels correlated with relative distance in both tasks, while orientation-related clusters (~500 and 700 ms) appeared only in the layout discrimination task, particularly around occipital channels at 700 ms.

Collectively, these findings reveal a clear temporal dissociation in MEG responses, with early distance coding followed by later orientation coding that parallels the spatial dissociation observed in fMRI. This temporal sequence suggests a hierarchical and dynamic organization of boundary feature representations in the human brain. The rapidly emerged relative distance encoding is consistent with early feature extraction that is largely automatic and task-invariant, as shown in previous studies on natural scene perception [33–37]. At the same time, the delayed emergence of orientation coding aligns with known feedback processing windows [35,38,39], implying that feedback-related processes may further refine orientation representations at later stages.

## Linking fMRI and MEG responses

To directly relate the spatial and temporal findings, we compared representational patterns across modalities. Group-averaged RDMs from V1 and three scene-selective areas in the fMRI experiment were partial correlated with MEG RDMs from occipital channels at each time point, without incorporating any feature-based model RDMs. This analysis aimed to identify shared representational patterns between hemodynamic and electromagnetic signals independent of specific encoding models.

The results showed that fMRI responses in V1 made a significant and sustained contribution to MEG responses in occipital channels across both tasks. In the layout discrimination task, significant correlations were observed from 70 to

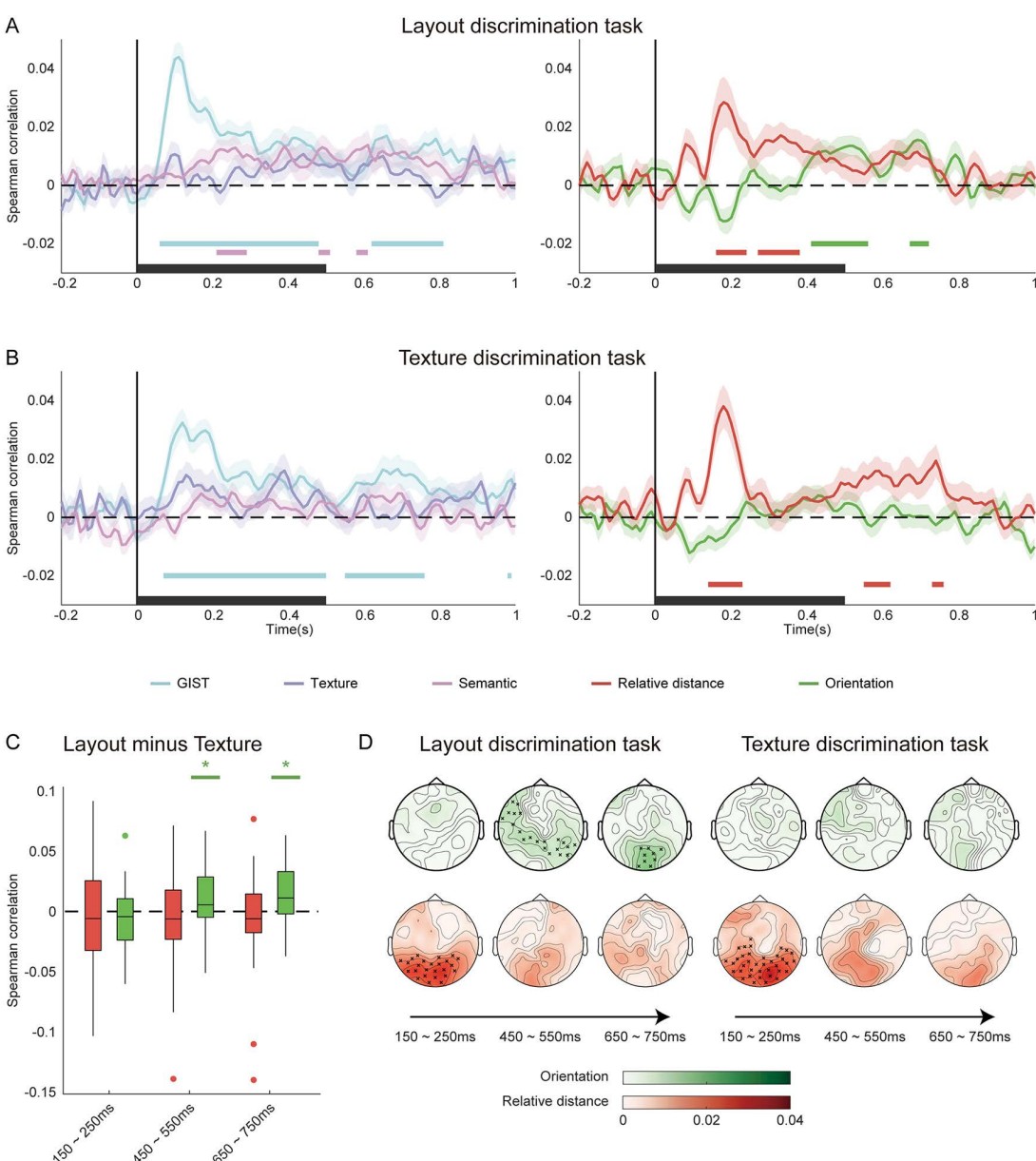

**Fig 7. Model representations for 3D layout: Matterport3D MEG experiment. A**, Layout discrimination task. **B**, Texture discrimination task. For **A** and **B**, significance was determined using one-sample t-tests against chance level, corrected for multiple comparisons with TFCE one-tailed $p < 0.05$. Colored horizontal bars indicate significant clusters, and black bars on the x-axis denote image presentation periods. **C**, Task-dependent enhancement of layout representation in three time windows. Asterisks denote significance in one-tailed $t$ test against chance level. * $q < 0.05$; ** $q < 0.01$, *** $q < 0.001$. **D**, Searchlight results. Scalp maps show partial correlations of the relative distance (red) and orientation (green) model RDMs with neural RDMs. Crosses mark significant clusters (TFCE one-tailed, $p < 0.05$). The scalp maps were generated using FieldTrip toolbox (https://www.fieldtriptoolbox.org/). The data underlying this figure can be found at https://doi.org/10.17605/OSF.IO/UXWR4.

840 ms (Fig 8A). In the texture discrimination task, three distinct clusters were identified (70–450 ms, 480–600 ms, and 630–850 ms; Fig 8B). Notably, direct task comparison revealed a marginally significant enhancement of V1–MEG correspondence in the 650–750 ms window ($t(31) = 2.75$, $q = 0.0587$, two-tailed, FDR corrected; Fig 8C).

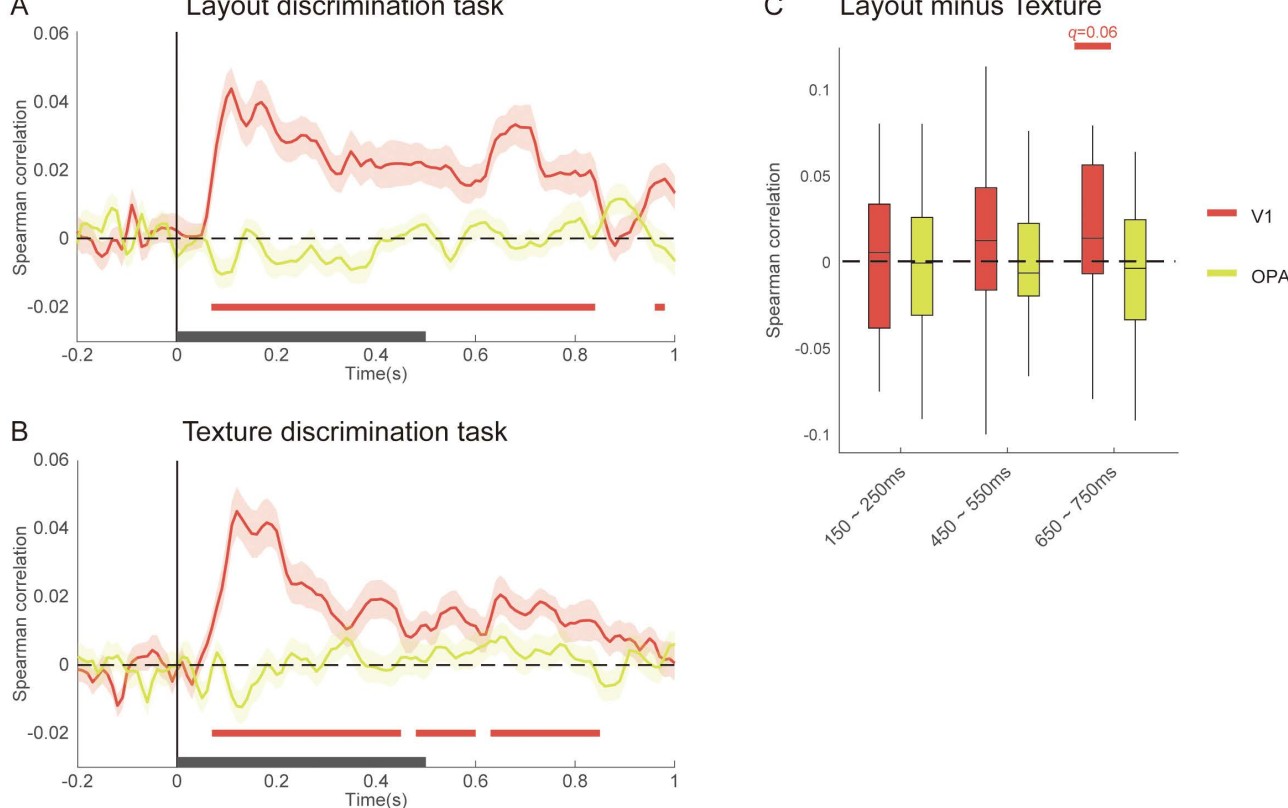

**Fig 8. Cross-modal representational similarity between fMRI and MEG.** Partial correlation between fMRI RDMs (V1, OPA) and the MEG RDMs from occipital channels for **A** layout discrimination task and **B** texture discrimination task. Significance was determined using one-sample t-tests against chance level, corrected for multiple comparisons with TFCE (one-tailed $p < 0.05$). Colored horizontal bars indicate significant clusters, and black bars on the x-axis denote image presentation period. **C**, Task-dependent changes in partial correlation across three critical time windows. The data underlying this figure can be found at https://doi.org/10.17605/OSF.IO/UXWR4.

This late enhancement coincides temporally with the orientation-specific feedback effects observed in the MEG analyses (Fig 7C and 7D), providing convergent evidence for navigation-related feedback to early visual cortex during active layout processing.

## Discussion

The present study investigated how the human brain represents critical 3D layout features in naturalistic indoor scenes. We combined large-scale fMRI analysis, based on a new computer vision-based reconstruction approach, with targeted fMRI and MEG experiments using ground-truth annotated natural scenes. This multimodal approach enabled us to characterize fine-grained neural representations of boundary-related information at both the spatial and temporal levels.

Our findings provide converging evidence for a spatiotemporal dissociation between boundary distance and orientation representations. Early visual areas primarily encoded the relative distance to boundaries, whereas scene-selective areas were more engaged in representing boundary orientation. Integrating fMRI and MEG results further revealed distinct temporal profiles underlying these two forms of spatial coding: distance encoding emerged rapidly and remained task-invariant, peaking around 200 ms after image onset; orientation encoding appeared later, reaching its peak after 400 ms,

and was strongly modulated by navigation-relevant task demands. Together, these results suggest a hierarchical organization of spatial layout representation along both cortical and temporal dimensions.

## Methodological advances

Previous studies on wall or boundary representation have faced two major methodological challenges that limit their ecological validity in the context of real-world navigation. First, most studies employed simplified, artificial stimuli consisting of isolated walls or geometric primitives. Although such stimuli facilitate precise experimental control, they lack critical real-world image regularities, such as semantic context, natural lighting conditions, and visual complexity [40]. Conversely, naturalistic scene images provide more realistic inputs but lack quantitative ground-truth 3D layout information. Moreover, artificial scenes typically contain few or no objects, allowing for unobstructed visibility of wall contours. In contrast, naturalistic images include numerous objects that fragment the visibility of wall boundaries, making edge-based wall segmentation particularly challenging. To overcome these gaps, our study developed a computer vision-based approach that accurately estimates boundary distance and orientation from natural indoor scenes, enabling their direct application to large-scale datasets such as the NSD. With modest annotation effort, this approach enables efficient testing and comparing 3D layout models across existing public neuroimaging datasets, thereby bridging the methodological divide between controlled and naturalistic paradigms.

Second, many previous studies relied on tasks that bore limited relevance to navigation, potentially underestimating top-down modulations in spatial representation. Growing evidence indicates that scene-selective areas are strongly influenced by navigational context [9,12,41,42]. For instance, Bonner and Epstein [9] showed that OPA can distinguish doors from shape-compatible paintings, while Aminoff and Tarr [41] demonstrated enhanced PPA and OPA activity when viewing scenes through windows versus picture frames. Our study directly builds upon these insights by incorporating tasks that vary in navigational relevance, thereby providing compelling evidence for task-dependent representation of 3D layout features.

While our current implementation focuses on indoor scenes, the camera calibration procedure is readily extendable to urban outdoor environments in which rectilinear structures (e.g., building, facades, and sidewalks) provide stable geometric cues. Its applicability is naturally more limited in environments that lacks orthogonal planar surfaces, such as natural landscapes or irregular architectural spaces. Moreover, the present layout reconstruction framework is strictly grounded in the Manhattan World assumption, which presumes mutually orthogonal planar structures. Extending the method to scenes with curved or non-orthogonal geometries would therefore require additional structural constraints and more sophisticated annotation procedures. By explicitly delineating these geometric requirements, our framework offers a practical bridge between tightly controlled experimental paradigms and naturalistic navigation research across a broad range of man-made environments.

## Implications in spatial layout processing

Inspired by the principle of vector coding [17], our study modeled egocentric boundary features in terms of their distance and orientation relative to the observer. A key finding is the clear dissociation between early visual areas and scene-selective areas in representing these two layout components, suggesting a hierarchical process for extracting spatial layout information from visual input.

Early visual areas exhibited robust sensitivity to the relative distance, even after controlling for 2D visual and semantic features. The fact that relative distance model captured the response pattern in these areas suggest that early visual cortex may play an essential role in estimating distance by segmenting the 2D image based on wall positions. This may explain the superior performance of our relative distance model in early visual areas compared with other alternatives (S2 Fig), as those models primarily focus on global features and lack egocentric directional information.

In contrast, orientation encoding likely involves more integrative computations. Representing wall orientation provides an efficient means to infer navigational structure, since side walls in indoor environments are typically orthogonal and

exhibit minimal curvature. Consistent with prior findings implicating scene-selective areas such as OPA in layout representation [6–8,12,15,16,43], our results demonstrate that boundary orientation constitutes the key 3D layout feature encoded in these areas. This interpretation is further supported by research on depth perception, which identifies V3B as a key site for integrating multiple depth cues—including linear perspective, shading, texture, and binocular disparity—into coherent 3D representations [44–48]. Given the anatomical and functional proximity between OPA and V3B [5], our findings suggest that OPA may similarly integrate diverse depth-related signals—including area-based distance cues computed in early visual cortex—into more abstract structural representations of 3D layout.

In classical depth-processing frameworks, early visual cortex primarily supports cue-specific encoding, whereas the integration of multiple cues into a unified 3D percept occurs in higher-level region V3B [47]. Within this framework, the task-dependent enhancement observed in early visual cortex in the present study may arise from at least two, potentially complementary mechanisms. First, it may reflect feedback modulation from higher-level scene-processing regions when layout information becomes behaviorally relevant. Depth structure has been shown to reshape activity patterns in early visual cortex, and such modulation is strongly influenced by attention and contextual feedback [49–51]. Under navigation-relevant demands, top-down signals may therefore amplify boundary-related features, increasing the precision of local spatial encoding.

Second, the enhancement may reflect the short-term maintenance of depth cue–related features within early visual cortex itself. Converging evidence indicates that early visual areas can sustain sensory-like representations during visual working memory and imagery [52–61]. Because navigation tasks require transient retention of boundary orientation and spatial structure, early visual cortex may function as a high-resolution representational buffer that maintains task-relevant depth cues over short intervals. Both feedback-driven gain modulation and local maintenance mechanisms would increase the strength and detectability of orientation-related representations under task-relevant conditions.

Nevertheless, we do not exclude alternative or complementary explanations. Covert attention to specific image locations or 2D orientations, as well as memory-related maintenance of spatially localized features, may also contribute to late activity patterns in V1. Future studies employing orientation-filtered natural images or more selective feature manipulations may help further disentangle these contributions.

### Neural representation of self-pose

By leveraging recovered camera parameters from the NSD dataset, we found that pitch angle, but not roll angle, was robustly encoded in early visual and scene-selective areas. Pitch angle is a critical parameter for perceiving and navigating within 3D environments, influencing both spatial updating and memory for scenes [62]. The present findings provide the first direct evidence that pitch angle is explicitly encoded in the human visual cortex.

The representation of roll angles was not observed in any of the examined brain regions. We suggest that two factors may contribute to this null effect. First, from a data distribution perspective, the variability of roll angles within the NSD indoor scene subset was substantially lower than that of pitch angles, which have reduced statistical sensitivity for detecting roll-related representations. Second, from an ecological standpoint, large variations in roll are uncommon in natural viewing behavior, as observers typically maintain the horizon approximately aligned with the retinal or image frame. In contrast, variations in pitch occur frequently during everyday perception when individuals adjust their viewpoint to objects located at different elevations relative to eye level. Consequently, the human visual system may be more attuned to changes in pitch-related changes than to roll during the natural scene processing.

### Absence of precise distance representation

Although scene-selective areas have been reported to encode distance and openness [10,11,13,16,63,64], our findings suggest that precise metric distance may not provide the most appropriate model for describing how wall distance is represented in the brain. While both models were designed to characterize aspects of wall distance, only the relative distance

model showed consistent and significant associations with neural activity across both full and partial correlation analyses (see S7 Fig for results of full Spearman correlation). This pattern suggests that although wall distance is encoded—particularly in the early visual cortex—the representation may rely more on relational, image-derived geometric cues than on absolute metric estimates. The absence of reliable effects for the precise distance model further implies that the visual system may preferentially employ intuitive, heuristic-based spatial representations grounded in visual geometry, such as the relative image area occupied by surfaces.

The absence of precise distance representation may also reflect the computational demands of the task. Estimating precise distance requires at least one known coordinate in the world coordinate system, such as the real-world size of an object or the camera's height. The brief static viewing (500 ms, or one saccade during natural movement) likely provides insufficient cues or processing time for such metric recovery. Future research employing dynamic naturalistic video with depth annotations may help clarify this issue.

## Conclusions

In summary, we developed quantitative methods for reconstructing 3D layout information from naturalistic indoor scene and used it to reveal the neural architecture supporting human spatial layout perception. Our findings demonstrate that the human visual system represents boundary distance and orientation through a hierarchically organized process: engaging early visual areas for egocentric distance estimation and scene-selective regions for orientation encoding. These representations are further modulated by navigational relevance, highlighting the dynamic interplay between perception and action in the human brain during spatial navigation.

## Methods

### NSD experiment

**Ethics statement.** Informed written consent was obtained from all participants, and the experimental protocol was approved by the University of Minnesota institutional review board.

**The Natural Scenes Dataset.** We used fMRI data from the NSD, which is described in detail by Allen and colleagues [24]. The dataset includes whole-brain fMRI measurements from 8 participants (6 females; age range: 19–32). Over the course of a year, each participant viewed 9,000–10,000 colored natural scene images across 30–40 scan sessions. The full image set consists of 73,000 images, but only a subset of 1,000 images was viewed by all participants. These images were sourced from the Microsoft Common Objects in Context (COCO) database [65] and were presented for 3 s, following a 1-s gap between trials. Each image was shown three times, resulting in approximately 30,000 trials per participant. In each trial, participants performed a recognition task, determining whether the image had been presented in any previous session. The images were displayed at a visual angle of 8.4° × 8.4°, with participants instructed to focus on a central fixation dot superimposed on the image.

**Stimuli.** *Indoor NSD images:* For our analysis, we selected indoor images from the NSD dataset using the COCO panoptic segmentation labels. These labels, designed for semantic segmentation in computer vision, delineate regions corresponding to both objects (thing) and background (stuff) within images. The panoptic labels consist of two main categories: "thing" (individual objects) and "stuff" (background elements like walls, floors, and ceilings). We focused on images that met the following criteria: (1) the image contained at least two of the three major layout components—ceiling, walls, and floor; (2) the walls were orthogonal to each other; and (3) no more than three side walls were present.

To ensure the walls were clearly visible in the selected images, we applied further manual screening to exclude images that met any of the following criteria: (1) blurred outlines of the background, (2) large objects or people obscuring the center of the image, or (3) incomplete enclosures (e.g., open or partially visible rooms).

***Reconstructing camera's internal and external parameters:*** The reconstruction of camera's parameters is based on the computation of vanishing points [66]. Six human annotators were recruited for the annotation task. Each

annotator labeled a subset (mean = 292) of the filtered images and each image had one annotation. As shown in [Fig 1A_i](), the annotators were instructed to label two pairs of lines in the selected images. Each pair of lines marks two clear edges that are parallel to the ground and to each other in the space shown in the image. The two pairs need to be orthogonal to each other. The line-pairs can be found at the edges of the walls, tiles on the floor, or any squared surface such as tables. The intersection of a line-pair is the vanishing point where the parallel lines converge in the 3D space. The straight line connecting the two vanishing points ($V_1$ and $V_2$) constitutes the horizon line $H$ of the space ([S1A Fig]()).

The camera's internal and external parameters can help to map 2D image coordinates to 3D world coordinates with the equation:

$$m = K \left[R \mid t\right] X$$

where $m$ is the point in the camera coordinate system, $X$ is the same point in the world coordinate system, $K$ is the internal parameters, and $\left[R \mid t\right]$ is the external parameters: $R$ is the rotation matrix and $t$ is the translation matrix of the camera. The internal parameters contain focal length, skewness, and principal point in the form:

$$K = \begin{bmatrix} f & \gamma & u_0 \\ 0 & f & v_0 \\ 0 & 0 & 1 \end{bmatrix}$$

where $f$ is the focal length, $\gamma$ is the skewness, and point ($u_0$, $v_0$) is the location of principal point. The external parameters contain the camera pose, including yaw, pitch, roll, and translation of the camera. The external parameters can be written as a 3 × 4 transformation matrix.

$$\left[R \mid t\right] = \left[r_1 \; r_2 \; r_3 \mid t\right]$$

where $r_1$, $r_2$, $r_3$ represent columns in the rotation matrix and $t$ represents the translation. To simplify, the present study took the skewness, yaw, and translation $t$ to be zero and considered the principal point to be located at the center of the image. With the simplification, the focal length can be calculated as:

$$h = \sqrt{\left\| \overrightarrow{O_h V_1} \right\| \left\| \overrightarrow{O_h V_2} \right\|}$$

$$f^2 = h^2 - \left\| \overrightarrow{OO_h} \right\|^2$$

where $O_h$ is the projection point of the principal point to the horizon line $H$ and $V_1$, $V_2$ are the two vanishing points ([Figs 1A_ii]() and [S1B]()).

Since we restricted the yaw angle to be zero, the pitch and roll angles can be calculated by the rotation matrix determined by the vanishing point $V_s$ where the parallel lines extending straight front converge. $V_s$ can be located as the intersection of the horizon line $H$ and the vertical line through the principal point $O$. The paired vanishing point $V_c$ of the direction of left and right can be easily located with the equation ([S1C Fig]()):

$$\left\| \overrightarrow{O_h V_c} \right\| = h^2 / \left\| \overrightarrow{O_h V_s} \right\|$$

Given the locations of $V_s$ and $V_c$ and the focal length, the external parameters can be calculated as:

$$r_1 = K^{-1}V_c / \left\|K^{-1}V_c\right\|$$

$$r_3 = K^{-1}V_s / \left\|K^{-1}V_s\right\|$$

$$r_2 = r_1 \times r_3$$

and the pitch and roll angles are

$$pitch = \tan^{-1}(r_3(1)/r_3(3))$$

$$roll = \tan^{-1}(r_1(2)/r_2(2))$$

***Reconstructing scene layout:*** Another human annotator was recruited for the annotation task and each image had one annotation. As shown in Fig 1A$_{iv}$, the annotator was told to label two lines based on the number of side walls present: (1) If the image contains three side walls, label the two lines on the middle side wall. (2) If there are fewer than three side walls, label the two lines on any side wall. The first line $L_1$ marks an edge of the wall that is parallel to the ground. The two endpoints of the first line need to be located at the corners between the annotated wall and the side walls. The second line $L_2$ marks the height of the annotated wall. The endpoints of the second line need to be located at the intersection lines between the annotated wall and the ceiling and the floor, respectively.

The reconstruction of scene layout needs to meet the prerequisites that all walls are orthogonal to each other and there are at most three side walls. Thus, all images that do not meet this prerequisite were discarded. In brief, the reconstruction includes the following steps: locating the third vanishing point, outlining the annotated wall, and outlining other side walls.

**Locating the third vanishing point.** The third vanishing point $V_3$ is the converging point of the parallel lines that are orthogonal to the ground. The principal point $O$ is the orthocenter of the triangle that is formed by $V_1$, $V_2$, and $V_3$. Thus, the third vanishing point $V_3$can be easily calculated (Figs 1A$_{iii}$ and S1D).

**Outlining the annotated wall.** According to the prerequisite, the conjunctions between the annotated wall and the ceiling and floor are parallel to the ground, and the conjunctions of the annotated wall and other side walls are orthogonal to the ground. The conjunctions of the annotated wall and the ceiling and floor intersect at the intersection point of the first line $L_1$ and the horizon line $H$. The conjunctions of the annotated wall and other side walls intersect at $V_3$. The detailed steps are following:

1. Locate the intersection point $V_a$ of $L_1$ and $H$.

2. Connect $V_a$ to the two endpoints of the second line $L_2$. These lines are the conjunctions of the annotated wall and the ceiling and floor.

3. Connect $V_3$ to the two endpoints of the first line $L_1$. These lines are the conjunctions of the annotated wall and other side walls. The intersection points of these two lines and the ceiling are named as $p_2$ and $p_3$. The intersection points of the two lines and the floor are named as $p_6$ and $p_7$.

The quadrilateral decided by $p_2$, $p_3$, $p_6$, and $p_7$ is the outline of the annotated wall (Fig 1A$_{iv}$).

**Outlining other side walls.** Other side walls are orthogonal to the annotated wall. The intersection point $V_o$ of other walls and $H$ follows the equation:

$$\left\|\overrightarrow{O_hV_o}\right\| = h^2/\left\|\overrightarrow{O_hV_a}\right\|$$

and $V_o$ thus can be easily located. We outlined other side walls with the following steps:

1. Locate the intersection point $V_o$ using the equation.

2. Connect $V_o$ to $p_2$ and $p_6$. These lines are the conjunctions of the left wall and the ceiling and floor. The intersection points of these two lines and the edges of the image are named as $p_1$ and $p_5$. If $p_2$ and $p_6$ are on the edges of the image, skip this step.

3. Connect $V_o$ to $p_3$ and $p_7$. These lines are the conjunctions of the right wall and the ceiling and floor. The intersection points of these two lines and the edges of the image are named as $p_4$ and $p_8$. If $p_3$ and $p_7$ are on the edges of the image, skip this step.

The area to the left of the polyline decided by $p_1$, $p_2$, $p_5$, and $p_6$ is the left wall. The area to the right of the polyline decided by $p_3$, $p_4$, $p_7$, and $p_8$ is the right wall (Fig 1A$_{v,vi}$).

**Orientation of side walls.** The orientation angle relative to the front is decided by the point $V_a$ and $V_o$.

$$\theta_a = \tan^{-1}(\left\|\overrightarrow{O_hV_a}\right\|/h)$$

$$\theta_o = \tan^{-1}(\left\|\overrightarrow{O_hV_o}\right\|/h)$$

where $\theta_a$ is the orientation angle of annotated wall and $\theta_o$ of orthogonal walls.

**Feature spaces.** *GIST:* The GIST feature quantifies spatial frequencies and orientations across different locations in a scene [25]. For each scene image, we first converted it to grayscale and then processed it using 512 filters (or channels), which consisted of 4 spatial frequencies and 8 orientations at 16 spatial locations (4 rows × 4 columns). To reduce the dimensionality of the resulting data, we performed principal component analysis (PCA) on all 73,000 NSD images. The top 59 principal components accounted for over 95% of the variance and were retained as the GIST feature space for each image.

*Texture:* To quantify the mid-level features of scene images, we used the texture model proposed by Portilla and Simoncelli [26], which has been shown to predict both human perceptual judgments and neural responses to textures [67,68]. First, each scene image was converted to grayscale and resampled to a resolution of 256 × 256 pixels. The images were then decomposed into orientation and spatial frequency sub-bands using the steerable pyramid transformation. These sub-bands represent spatial maps of complex-valued coefficients, where the real part corresponds to the response of V1 simple cells and the magnitude of the coefficients corresponds to the response of V1 complex cells.

The model computed several descriptors within and across these sub-bands to capture texture features. Specifically, the descriptors included: (1) pixel autocorrelations at different positions within each sub-band, which characterize periodicity; (2) magnitude autocorrelations and cross-correlations across sub-bands, which capture structural elements like edges and corners, as well as second-order texture properties; and (3) cross-correlations of the real part of the sub-bands with phase-doubled responses at the next coarser scale, which distinguish edges from lines and capture gradients of shading.

 

Each scene image was processed through the texture model using 4 orientations and 4 spatial frequencies across a 5×5 neighborhood at each of the 4 (2 rows×2 columns) spatial locations. We performed principal component analysis (PCA) on the texture parameters derived from all 73,000 NSD images. The top 125 principal components accounted for over 95% of the variance and were retained as the texture feature space for each image.

*Semantic:* To model the semantic properties of the scene images, we employed the object2vec model [27], which is based on word2vec, a well-established algorithm in computational linguistics used to model the lexical-semantic properties of words based on their co-occurrence statistics [69]. Object2vec extends this idea to image semantics by learning to represent image labels in a continuous vector space. Specifically, the method uses the continuous bag of words (CBOW) model of word2vec, where the target label is predicted based on the surrounding context labels within an image.

In this study, we trained the object2vec model on the training set of the COCO database, which contains 118,287 images. For training, each image was converted into a "sentence" consisting of its panoptic segmentation labels. The panoptic segmentation labels in COCO are divided into 80 "thing" categories (e.g., objects such as people, cars) and 53 "stuff" categories (e.g., background regions like walls, sky), totaling 133 labels.

The object2vec embeddings were learned by training the CBOW model of word2vec with an embedding dimension of 50. The model was trained for 100 epochs, using a negative sampling parameter of 20 and a window size of 15, meaning that the model would consider 30 surrounding labels within each image to predict the target label. After training, the object2vec model generated a 50-dimensional vector to represent each label. The semantic feature space for each image was obtained by averaging the vectors of all its labels, providing a compact representation of its semantic content.

*Layout orientation model:* The layout orientation model represents the orientations of the side walls (i.e., the boundaries) in a scene (Fig 2C, left). Using the aerial view annotations, we calculated the orientations and start and end points of the side walls. For each side wall, we computed its normal vector—unit vector that is perpendicular to the wall and directed towards the camera. These normal vectors are represented in a right-handed 2D coordinate system, with the +*x* axis pointing to the right and the +*y* axis pointing forward.

To discretize the orientation information across different locations, we quantized the normal vectors into 5 histogram bins. These bins are organized along the horizontal axis in the perspective of observer, evenly dividing the entire field of view. The orientation information of the side walls is integrated across these location bins using the following equations:

$$F_i(\phi) = max\left(0, \frac{\sigma - abs(\phi - \phi_i^c)}{\sigma}\right)$$

$$R_i = \int O(\phi)F_i(\phi)d\phi$$

where $\sigma$ is the angular width of each bin, $\phi_i^c$ is the center of the *i*-th bin, and $\varphi$ represents any direction in the field of view. The responses of the bins are computed by integrating over the entire field of view, where $F_i(\phi)$ is the soft-histogram function for the *i*-th bin, and $O(\phi)$ corresponds to the orientation angle of the wall at the direction $\varphi$.

*Layout relative distance model:* Assuming that all walls are orthogonal to each other, the proportion of the side wall relative to the entire pixel space provides a rough estimate of the relative distance of the side walls. As the proportion of side wall pixels increases, the perceived distance to the side walls decreases.

In the relative distance model (Fig 2C, right), each scene image is divided into 5 equal horizontal bins. For each bin, the proportion of side wall pixels is calculated based on the layout segmentation map in the pixel space. This proportion serves as a coarse measure of the relative distance of the side walls within each bin.

**MRI data acquisition and preprocessing.** Functional data were acquired at 7T using whole-brain gradient-echo echo-planar imaging (EPI) with an isotropic resolution of 1.8 mm and a repetition time (TR) of 1.6 s. The functional data

were preprocessed by performing temporal interpolation to correct for slice timing and spatial interpolation for head motion correction. A General Linear Model (GLM) approach was applied to estimate single-trial beta weights. The third beta version (betas_fithrf_GLMdenoise_RR) was used in this study. In brief, a new GLMsingle algorithm [70] was used to estimate voxel responses. This algorithm implements optimized denoising and regularization procedures to accurately measure changes in brain activity evoked by experimental stimuli. For further details on this methodology, please refer to Allen and colleagues [24].

**Defining ROIs.** The selection of regions of interest (ROIs) was informed by prior research on visual navigation [7,9,12,15,16], focusing on both early visual areas and scene-selective regions. Specifically, retinotopic and scene-selective ROIs were defined based on functional localizers from the NSD experiment. The category localizer task was employed to identify scene-selective areas, including the PPA, OPA, and RSC, as well as the face-selective fusiform face area. A pRF (population receptive field) mapping task was used to define the retinotopic visual area V1.

**Decoding analysis.** For each participant, a support vector regression (SVR) model was trained to decode the neural representation of pitch and roll angles. SVR is a machine learning algorithm commonly used for regression analysis and is particularly effective in decoding continuous variables [71]. The epsilon SVR was implemented using LibSVM with a linear kernel. A 10-fold cross-validation procedure was employed to assess the performance of the SVR model. Decoding performance was quantified by computing the Pearson correlation between the predicted and actual angles. The chance level was defined as zero, indicating no correlation between the predicted and actual angles. Group-level statistical analysis was conducted using a one-tailed $t$ test against the chance level. False discovery rate (FDR) correction was applied to adjust for multiple comparisons across ROIs.

**Representational similarity analysis.** Neural RDMs were constructed within each ROI for each participant. Each scene image was presented up to three times. Multivoxel activation patterns for each image were obtained by averaging all trials of the image. The distances between resulting patterns were calculated using Pearson correlation, which was then used to construct the RDMs.

Model RDMs were generated for all the feature spaces described previously. The RDMs of GIST and texture features were calculated using Euclidean distance, while the RDM of semantic features was measured using cosine distance. The RDMs of the layout orientation and relative distance models were measured using city block distance.

To identify the neural representation of the feature spaces, we compared the neural and model RDMs for each participant. A partial correlation analysis was applied to characterize the coding of layout, while controlling for perceptual and semantic scene features. We incorporated the models of layout orientation and relative distance, while controlling for GIST, texture, and semantic features. One-tailed t-tests were used across participants to assess the partial effects for statistical significance, with FDR correction applied for multiple comparisons across all models and ROIs.

**Searchlight analysis.** The searchlight analysis was conducted in the native space of each participant with the mask comprised of "nsdgeneral," "floc-places," "floc-faces," and "floc-bodies" provided in the NSD dataset [24]. The mask covered voxels responsive to the NSD experiment in the posterior aspect of cortex and fROIs in the native space. We moved a cubic searchlight in the radius of 5 mm in the mask. At each voxel, we obtained the partial correlations of orientation and relative distance models while controlling for image features. The searchlight results were transformed to surface space for visualization using the interpolation of nearest-neighbor.

## Matterport3D fMRI experiment

**Ethics statement.** All participants provided written informed consent before the experiment and were compensated for their time. The study was approved by the Committee for Protecting Human and Animal Subjects at the School of Psychological and Cognitive Sciences at Peking University (Institutional Review Board Protocol No: 2023-08-03).

**Participants.** Thirty healthy adults (mean age = 21.53 years, SD = 2.16; 11 females) with normal or corrected-to-normal vision participated in the study.

**Stimuli.** *Naturalistic scene images:* We selected 60 naturalistic indoor images from the Matterport3D database, a large-scale RGB-D dataset comprising 10,800 panoramic views generated from 194,400 RGB-D images across 90 building-scale environments [32]. The database provides depth map, camera pose, and surface normal map for each image. Additionally, ground-truth layout annotations were sourced from the Matterport3D-Layout dataset, a specialized database for scene layout estimation [72].

The images were categorized into five indoor scene types: bathroom, bedroom, kitchen, living room, and hallway. To maintain consistency, all selected images had a pitch and roll angle of 0° and were extracted from panoramic views with a horizontal field of view (FOV) of 60° and a vertical FOV of 53.3°. The final images were resized to 650 × 520 pixels for experimental use.

*Synthesis of layout segmentation map:* We generated a 'ground-truth' segmentation map along with two additional segmentation maps, each incorporating a 10° yaw angle deviation to the left and right relative to the original image.

The process began by reconstructing an aerial view map of the original image using the depth and normal maps provided by the Matterport3D database. Based on this reconstructed aerial view, we generated aerial view maps with yaw angle deviations of ±10°. Given these aerial view maps and the ceiling height information from the Matterport3D database, the segmentation maps were synthesized using the method described in the Reconstructing Scene Layout section of the NSD experiment.

Consistent with the labeling approach in the NSD experiment, we assigned labels to the side walls in the layout segmentation maps. The side wall regions were labeled sequentially from left to right.

*Synthesis of texture image:* For each image, we synthesized both an original texture image and a feature-swapped texture image.

The original texture images were generated to match a set of descriptors derived from the texture model used in the NSD experiment for the original image. Specifically, the model's responses to the original image were computed using four orientations and four spatial frequencies across a 7 × 7 neighborhood at each of the nine spatial locations (3 rows × 3 columns). An image of Gaussian white noise was then iteratively adjusted to match these model responses. This synthesis procedure approximates sampling from the maximum entropy distribution of images that conform to a given set of model responses. To generate the texture images, we applied gradient descent for 50 iterations.

To synthesize the feature-swapped texture image, we replaced the pixel autocorrelation descriptor of the original image with that of a randomly selected image from the Matterport3D database, ensuring that this selected image had not been presented in the task. The same iterative procedure was then applied to match the swapped model responses.

**Feature spaces.** We employed the GIST, texture, and semantic models to quantify image features. The GIST and texture features were computed as the NSD experiment using same model parameters. The responses of texture models were estimated among all scene images with a pitch angle of 0° in the Matterport3D database, in total of 64,800 scene images. Then we performed the PCA on the responses of texture models. The top 118 principal components accounted for over 95% of the variance and were retained as the texture feature space for each image. The responses of GIST model were directly used without PCA. The semantic feature was computed by the panoptic labels rendered from the 'house segmentations' in the Matterport3D database using the camera parameters. An object2vec model was performed as the NSD experiment using the same model parameters except for an embedding dimension of 20. For layout representation, we incorporated the two layout models described in the NSD experiment. The relative distance model was computed from the ground-truth layout segmentation map in the Matterport3D-Layout dataset and the orientation model was computed from the aerial view map, reconstructed using the segmentation, depth, and normal maps.

**Experimental design.** All experiments were conducted using Psychtoolbox software. Stimuli were displayed on an LCD monitor (refresh rate, 60 Hz; resolution, 1,920 × 1,080), subtending a visual angle of 9.5° × 7.6°. Participants viewed the stimuli at a distance of 182 cm through a mirror placed above their eyes. Each participant completed two scanning

sessions, with each session consisting of three runs of the layout discrimination task and three runs of the texture discrimination task. Additionally, two localizer runs were performed before task runs in the first session.

*Layout discrimination and texture discrimination tasks:* We employed a match-to-sample paradigm for both tasks (Fig 4A). Each trial commenced with a fixation cross displayed for 1–3 s, followed by a scene image presented for 0.5 s. After an additional fixation period of 3–5 s, the response phase began, during which two stimuli appeared on the left and right sides of the screen. Participants were instructed to select the stimulus that best matched the initial scene image by pressing a button using their right hand. The response window lasted 3.5 s; however, the stimuli disappeared once a response was made. Inter-trial intervals were jittered and pseudo-randomized. To ensure an adequate baseline signal, two 16-s rest periods were included at the beginning and end of each run. Each run comprised 40 trials, totaling 482 s in duration.

In the layout discrimination task, participants were presented with two layout segmentation maps during the response phase: one representing the "ground-truth" layout of the scene and another with a deviation. Participants were instructed to choose the map that accurately depicted the scene's layout. In the texture discrimination task, participants were shown two texture images: one corresponding to the original scene and another feature-swapped version. They were asked to select the image that best matched the scene's color distribution.

Within each session, participants completed all 3 runs of one task before proceeding to the next task, with the order of tasks counterbalanced across sessions. Each scene image was presented twice per task in a session, ensuring no image was repeated within a single run.

*Functional localizer runs:* We employed a block design to define the functional regions of interest (fROIs) for each participant. The localizer stimuli consisted of colored images from four categories: scenes, faces, objects, and scrambled objects. Images within the same category were presented in centrally displayed blocks lasting 16 s. Each block contained 20 images, with each image presented for 300 ms, followed by a 500 ms blank interval. Inter-block intervals lasted 8 s.

Participants performed a one-back task, in which they were required to press a button when two consecutive images were identical. Each localizer run lasted 408 s and comprised four blocks per category. The order of category blocks was counterbalanced across participants and runs.

**MRI data acquisition.** BOLD fMRI data were collected using a 3T Siemens Prisma scanner equipped with a 64-channel receiver head coil. Functional images were acquired using a multiband EPI sequence with the following parameters: multiband factor = 2, repetition time (TR) = 2 s, echo time (TE) = 30 ms, matrix size = 112 × 112 × 62, flip angle = 90°, spatial resolution = 2 × 2 × 2.3 mm$^3$, and number of slices = 62.

Before the functional scans in each session, a high-resolution T1-weighted 3D anatomical dataset was acquired for each participant to aid in image registration. The anatomical scans were obtained using an MPRAGE sequence with the following parameters: TR = 2,530 ms, TE = 2.98 ms, matrix size = 448 × 512 × 192, flip angle = 7°, spatial resolution = 0.5 × 0.5 × 1 mm$^3$, number of slices = 192, and slice thickness = 1 mm.

**fMRI pre-processing.** The anatomical and functional data were pre-processed and analyzed using AFNI [73]. Functional images were slice-time corrected using the AFNI function 3dTshift and motion-corrected to the reference image, which was considered the minimum outlier (using 3dVolreg). The functional images from each session were aligned with the corresponding anatomical images. The functional data from the second session were then co-registered and resampled to match the space of the first session, guided by the alignment of the two anatomical images. After co-registration, the signal amplitudes of the functional images were rescaled to a 0–200 range. All RSA analyses were conducted in the original space.

The responses from the task runs were modeled using a GLM. Each trial was modeled with a canonical Hemodynamic Response Function (HRF), derived from the onset of the original image, and convolved with a 0.5-s square wave. Additional regressors included the participant's response, six motion parameters, and three polynomial terms to account for

slow signal drifts. We performed the least-squares-sum estimation within AFNI to obtain single-trial beta estimates [74]. These single-trial beta values were then normalized within each run for subsequent analyses.

**Defining ROIs.** For the Matterport3D fMRI experiment, ROI selection was likewise guided by established findings in visual navigation and aligned with our NSD experiment, focusing on both early visual areas and scene-selective regions. However, because retinotopic mapping was not performed in this experiment, we defined a representative early visual area (V1) with a conservative anatomical-parcellation approach.

Using the functional localizer runs, we defined three scene-selective ROIs in each hemisphere: the PPA, the OPA, and the RSC. We fitted the response model using a GLM in AFNI (3dDeconvolve and 3dREMLfit). BOLD responses were modeled by convolving a standard HRF with a 16-s square wave for each category. Estimated motion parameters, participants' responses, and three polynomial terms to account for slow drifts were included as regressors of no interest.

Scene-selective areas were defined as contiguous clusters of voxels with a threshold of $p < 1 \times 10^{-4}$ (uncorrected) under the contrast of scene > face, based on their anatomical locations. Specifically, the PPA was defined by locating the cluster between the posterior parahippocampal gyrus and the lingual gyrus, the OPA was defined near the transverse occipital sulcus, and the RSC was located near the posterior cingulate cortex. In seven participants, the localizer failed to identify the RSC. The primary visual cortex (V1) was defined based on each participant's anatomical parcellation from Freesurfer [75].

**Representational similarity analysis.** For each participant, neural RDMs were constructed for each task within each ROI. Each image was presented four times per task. Multivoxel activation patterns for each image were derived by averaging the trials across sessions. The distances between patterns were calculated as one minus the Pearson correlation.

Model RDMs were constructed for the image feature models, layout orientation model, and layout relative distance model using the same distance measure as in the NSD experiment.

The partial correlation analysis, similar to those in the NSD experiment, was performed separately for the two tasks to characterize the task-dependent representations of layout. We included the models of layout orientation and layout relative distance in the analysis, again controlling for the three image feature models. One-tailed t-tests were applied across participants to assess the statistical significance of partial effects, with FDR correction for multiple comparisons across all models and ROIs. The full Spearman correlation results for the layout models are shown in S7 Fig.

## Matterport3D MEG experiment

**Ethics statement.** All participants provided written informed consent before the experiment and were compensated for their time. The study was approved by the Committee for Protecting Human and Animal Subjects at the School of Psychological and Cognitive Sciences at Peking University (Institutional Review Board Protocol No: 2024-10-02).

**Participants.** Thirty-two healthy adults (mean age = 21.22 years, SD = 2.51; 20 females) with normal or corrected-to-normal vision participated in the study.

**Stimuli.** The same set of 60 indoor scene images, along with their synthesized segmentation maps and texture images, were used in the MEG experiment as in the fMRI experiment. The stimuli were displayed on a rear-projection screen (refresh rate: 60 Hz) with a visual angle of 9.5° × 7.6°. Participants viewed the stimuli at a distance of 91 cm.

**Experimental design.** The design of the MEG experiment closely followed the match-to-sample paradigm from the fMRI experiment. Each trial began with a fixation period of 1–3 s, followed by the presentation of a scene image for 0.5 s. After another fixation period lasting 3–5 s, the response phase began, during which two stimuli were presented side-by-side on the screen. Participants were instructed to select the stimulus that best described the previously shown image within 3 s using their right hand.

Both the layout discrimination task and the texture discrimination task consisted of three runs each. All 60 scene images were presented in each run, with each image shown three times per task. All tasks were completed within a single session.

**Data acquisition and preprocessing.** Electromagnetic brain activity was recorded using an Eekta Neuromag 306 MEG system, which consists of 204 planar gradiometers and 102 magnetometers. The system includes 102 triple-sensor elements, with each sensor comprising one magnetometer and two planar gradiometers. Data were sampled continuously at 1,000 Hz and band-pass filtered online between 0.1 and 330 Hz.

Offline preprocessing was performed using the MNE-Python package. Temporal signal space separation was applied to reduce environmental and head motion artifacts. Independent component analysis (ICA) was then used on the data with the fastICA algorithm implemented in MNE-Python. Components associated with eye blinks and saccades were identified and removed from the raw unfiltered data. Subsequently, the data were demeaned, detrended, and down-sampled to 100 Hz. A time window of 1,700 ms was applied to segment the raw data, spanning from 200 ms before the first image onset to 1,500 ms after the onset. Trials were band-pass filtered between 0.1 and 30 Hz and then excluded if the gradiometer value exceeded 5,000 fT/cm or if the magnetometer value exceeded 5,000 fT.

**Representational similarity analysis.** RSA in Matterport3D MEG experiment was conducted using the CoSMoMVPA toolbox. The event-related RSA was performed across 24 occipital magnetometers and one of their combined gradiometers. The selection of these channels was based on the extensive involvement of occipital regions in encoding layout, as observed in the previous two fMRI experiments. Temporal smoothing was applied by averaging over two adjacent time points (±20 ms). Neural RDMs were constructed at each time point for each task and participant. For each image, all trials were averaged, and distances between images were calculated as one minus the Pearson correlation coefficient.

The partial correlation analysis conducted was performed across all time points within the trial. The analysis incorporated the model RDMs of three image features, layout orientation, and layout relative distance. The full Spearman correlation results for the layout models are presented in S7 Fig.

To compare the correspondence between the representations in fMRI and MEG, we computed group-level fMRI-RDMs by averaging the RDMs of all participants for V1 and three scene-selective areas, respectively. Partial correlation analysis was then applied to compare the fMRI-RDMs of V1 and OPA with the MEG-RDMs of each participant at each time point.

Statistical significance was assessed against chance levels by computing random-effect temporal-cluster statistics, corrected for multiple comparisons. The null distribution was generated through t-tests over 10,000 iterations, in which the sign of decoding performance above chance was randomly flipped. Threshold-free cluster enhancement (TFCE) was used as the cluster statistic [76], with a threshold step of 0.1. Significant temporal clusters were identified using a cluster-forming threshold of $p < 0.05$, one-tailed.

Time window-based RSA was performed across occipital magnetometers and one of their combined gradiometers. Here, we chose three 100-ms time windows corresponding to the significant clusters of layout models, centered at 200, 500, and 700 ms after image onset. All time points within each time window were averaged for each channel. The neural RDMs were constructed at each time window for each task and participant. The partial correlation analyses of model RDMs and fMRI-RDMs were conducted as the event-related RSA.

**Searchlight analysis.** The time window-based RSA method described above was applied in a searchlight approach across the entire scalp of each participant. All magnetometers and one of the combined gradiometers were involved in the searchlight analysis, resulting in 204 channels. For each channel, all time points within each time window were averaged. For each time window, task, and participant, the searchlight analysis was performed with a channel neighborhood determined by Delaunay triangulation. On average each location had a neighborhood of 16 channels. The same statistic test and TFCE procedures as described in the event-related RSA were applied to the partial correlation results across the scalp.

## Supporting information

**S1 Fig. The detailed illustration for calculating internal and external parameters.** The description for the procedures of the calculation can be found in Methods of the main text.
(DOCX)

**S2 Fig. The partial correlation analyses of alternative layout models for the NSD experiment.** Each plot shows the RSA results after adding one of the alternative models (mean depth model, 3D surface model, and fwall model), with the alternative model's partial correlation represented by the yellow box. **A**, Mean depth model. **B**, 3D surface model (surface orientation × distance) [16]. **C**, fwall model [15]. Asterisks denote significant results in the one-tailed $t$ test against chance level. * $q < 0.05$; ** $q < 0.01$, *** $q < 0.001$. The data underlying this figure can be found at https://doi.org/10.17605/OSF.IO/UXWR4.
(DOCX)

**S3 Fig. Representations of layout in other seven NSD participants.** Searchlight-based partial correlation was performed and the results are displayed on the flattened cortical surfaces.
(DOCX)

**S4 Fig. The partial correlation analyses of alternative layout models in the Matterport3D fMRI experiment. A and C**, RSA results including the precise distance model in layout and texture discrimination tasks, respectively. **B and D**, RSA results including the mean depth model in layout and texture discrimination tasks, respectively. Asterisks denote significant results in the one-tailed $t$ test against chance level. * $q < 0.05$; ** $q < 0.01$, *** $q < 0.001$. The data underlying this figure can be found at https://doi.org/10.17605/OSF.IO/UXWR4.
(DOCX)

**S5 Fig. The partial correlation analyses of alternative 2D feature models in the Matterport3D fMRI experiment. A and C**, RSA results including the central contrast model in layout and texture discrimination tasks, respectively. **B and D**, RSA results including the object clutter model in layout and texture discrimination tasks, respectively. Asterisks denote significant results in the one-tailed $t$ test against chance level. * $q < 0.05$; ** $q < 0.01$, *** $q < 0.001$. The data underlying this figure can be found at https://doi.org/10.17605/OSF.IO/UXWR4.
(DOCX)

**S6 Fig. The partial correlation analysis of alternative fixation model in the Matterport3D MEG experiment, incorporating the within-task farthest fixation patterns.** The 2D visual, texture and semantic models were also involved in this partial correlation analysis but are not displayed for clarity. Statistical testing procedures are consistent with those in Fig 7 of the main text. The data underlying this figure can be found at https://doi.org/10.17605/OSF.IO/UXWR4.
(DOCX)

**S7 Fig. The full Spearman correlation analysis of 3D layout models in Matterport3D fMRI and MEG experiments.** The left plots show full correlations between 3D layout models and ROIs RDMs in Matterport3D fMRI experiment. Statistical testing procedures of left plots are consistent with those in Fig 6 of the main text. The right plots show full correlations between 3D layout models and occipital channel RDMs in Matterport3D MEG experiment. Statistical testing procedures of right plots are consistent with those in Fig 7 of the main text. **A**, Layout discrimination task. **B**, Texture discrimination task. **C**, Task-dependent enhancement of representation. The data underlying this figure can be found at https://doi.org/10.17605/OSF.IO/UXWR4.
(DOCX)

**S1 Table. Partial correlations between 2D model RDMs and neural RDMs in layout discrimination task: Matterport3D fMRI experiment.**
(DOCX)

**S2 Table. Partial correlations between 2D model RDMs and neural RDMs in texture discrimination task: Matterport3D fMRI experiment.**
(DOCX)

**S3 Table. Task-dependent enhancement of representation, calculated as the difference between partial correlation coefficients for the two tasks (layout minus texture): Matterport3D fMRI experiment.**
(DOCX)

**S1 Text. Supplementary Methods and Results.**
(DOCX)

## Author contributions

**Conceptualization:** Yichen Wu, Sheng Li.

**Data curation:** Yichen Wu.

**Formal analysis:** Yichen Wu.

**Funding acquisition:** Sheng Li.

**Investigation:** Yichen Wu.

**Methodology:** Yichen Wu.

**Project administration:** Sheng Li.

**Software:** Yichen Wu.

**Supervision:** Sheng Li.

**Visualization:** Yichen Wu.

**Writing – original draft:** Yichen Wu.

**Writing – review & editing:** Yichen Wu, Sheng Li.

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
