## [Editor Report · Decision Letter 0]

20 Oct 2025

Dear Dr Li,

Thank you for submitting your manuscript entitled "Vectorial boundary representation of 3D layout in human visual cortex" for consideration as a Research Article by PLOS Biology.

Your manuscript has now been evaluated by the PLOS Biology editorial staff, as well as by an academic editor with relevant expertise, and I am writing to let you know that we would like to send your submission out for external peer review.

Once your full submission is complete, your paper will undergo a series of checks in preparation for peer review. After your manuscript has passed the checks it will be sent out for review. To provide the metadata for your submission, please Login to Editorial Manager (https://www.editorialmanager.com/pbiology) within two working days, i.e. by Oct 22 2025 11:59PM.

Kind regards,

Taylor

Taylor Hart, PhD,

Associate Editor

PLOS Biology

thart@plos.org

---

## [Decision Letter · Decision Letter 1]

4 Dec 2025

Dear Dr Li,

Thank you for your patience while your manuscript "Vectorial boundary representation of 3D layout in human visual cortex" was peer-reviewed at PLOS Biology. It has now been evaluated by the PLOS Biology editors, an Academic Editor with relevant expertise, and by several independent reviewers.

In light of the reviews, which you will find at the end of this email, we would like to invite you to revise the work to thoroughly address the reviewers' reports.

As you will see, the reviewers praised your approach and called the study well-executed. However, they also raised concerns about unaddressed confounds, as well as insufficient theoretical grounding and justification for some of the analytical choices, and missing information. In your revision, you should ensure that you address the concerns related to eye movement, the relevance of late V1 engagement for orientation, and possible confounds between orientation and relative distance in the partial correlation. In addition, we encourage you to expand on the points raised by Reviewer 3 about the applicability of your approach to outdoor spaces, as well as the distinct representations of self pose vs 3D layout, although we do not require you to perform further analyses to address these. You should also provide a thorough response to all of the reviewers' points.

Given the extent of revision needed, we cannot make a decision about publication until we have seen the revised manuscript and your response to the reviewers' comments. Your revised manuscript is likely to be sent for further evaluation by all or a subset of the reviewers.

**IMPORTANT - SUBMITTING YOUR REVISION**

*Re-submission Checklist*

*Published Peer Review*

*PLOS Data Policy*

*Blot and Gel Data Policy*

Sincerely,

Taylor

Taylor Hart, PhD,

Associate Editor

PLOS Biology

thart@plos.org

REVIEWS:

Reviewer #1: This paper reports three experiments investigating the neural correlates of scene layout, specifically: the orientation and distance of walls in pictures of natural scenes. The first experiment is a reanalysis of the NSD fMRI dataset. This is followed by task-based fMRI and MEG experiments. The main finding is a dissociation between the neural correlates of wall orientation and wall distance, with distance most strongly represented in early visual cortex (fMRI), early in time (MEG), and irrespective of task (navigation vs texture match-to-sample task), and orientation most strongly represented in scene-selective regions, with the early visual cortex activated late in time (MEG) and only in the navigation task.

This is an impressive body of work, with advanced models used to estimate various properties from natural scenes, state-of-the art neuroimaging analyses, and a nicely converging set of results from multiple methods and tasks. The paper is also well written and easy to follow. However, the interpretation of the results is not straightforward, particularly pertaining to the role of the early visual cortex (V1) in encoding navigation-relevant information (over and beyond its response to the visual image and modulations thereof).

Main comments:

-The authors control for 2D image features using various models. However, these models (e.g., GIST) are far from perfect and it is likely that there are residual visual differences that were not accounted for. Indeed, the variables of interest (e.g., wall distance and orientation) are themselves defined visually (e.g., orientation, known to be encoded in V1), such that fully accounting for visual feature differences would presumably eliminate any effect of interest? By controlling for image features, would you argue that the results do not reflect the processing of image features then? Finally, if the goal is to regress out image features, did you consider creating RDMs using feature representations in the various layers of DNNs (e.g., AlexNet), which should better capture visual features than GIST etc.?

-There is a close link between eye movements and fMRI responses to natural scenes (e.g., O'Connell & Chun, Nat Comm 2018). It was not clear if/how eye movements were controlled in the present work. It is quite possible that participants fixated different parts of a scene depending on its layout (e.g., near vs far). This is particularly concerning for the NSD study, where scenes were presented for 3 seconds. The NSD does include some eye tracking sessions - perhaps these can be analyzed to exclude this. Alternatively, you could run a separate eye tracking version of the task-based fMRI experiment to test whether eye movements differed systematically between conditions (these data could then be used to create an RDM based on eye movements and include this RDM in the fMRI analyses as a control model).

-One of the main results is the late and task-dependent representation of wall orientation in early visual cortex (V1). The late emergence of this response shows that it is not driven by the initial feedforward sweep but likely reflects an influence of spatial / feature-based attention or working memory (or eye movements, see above). If so, I find it hard to understand how V1 specifically "encodes" navigation-relevant properties (as the manuscript states) - that would be more a function of scene-selective regions. Rather, this could be more parsimoniously explained by a task-specific modulation of known V1 encoding properties: Attention to (or memory of) task-diagnostic locations in space and/or to orientations in the image could explain these findings. For example, to perform the task, participants would attend to wall orientation, and this is then reflected in V1 activity patterns, reflecting activity in orientation-selective neural populations.

Other comments:

-Did some of the dimensions (e.g., wall distance) covary with other picture properties, such as depth-of-field?

-The reported results are partial correlations; it would be relevant to know the correlations between the RDMs that are included in the analyses (e.g., GIST, texture, distance, orientation). This could be in the form of a correlation matrix, possibly in the Supp Material. For example, in Fig 7A, the negative correlation for orientation peaks at the same time (around 200 ms) as the positive correlation for distance, suggesting that these two models were related and that the negative correlation could be an artifact of the partial correlation method.

Reviewer #2: The manuscript asks how the human visual system represents navigation-relevant 3D layout from natural indoor images. The authors build a computer-vision pipeline that reconstructs wall distance and orientation (plus self-pose) from images, apply it to NSD fMRI, and validate with new fMRI/MEG tasks. Using RSA with controls for low/mid/high-level visual and semantic features, they report a dissociation: early visual cortex tracks relative boundary distance, while scene-selective areas (OPA/PPA) preferentially encode boundary orientation. MEG timing suggests distance emerges ~200 ms, task-invariant, whereas orientation peaks >400 ms and is boosted by navigation demands—with late feedback to early visual cortex

The paper is solid and well executed, but conceptually it extends an active line of work on scene layout and navigational affordances. I wonder whether the paper might be better suited for a strong field journal in cognitive neuroscience.

Major concerns:

1. The relative-distance proxy is derived from area differences (not metric distance). The authors should quantify how this proxy relates to true distance on Matterport3D and show that results hold with alternative formulations of distance to demonstrate that the results are truly distance -related. Since there is no result with metric distance, I'm worried that the effects in V1 can be due to other features that covary with the Distance measured as area difference. For instance, contrast in foveal area? Maybe smaller walls are more cluttered by objects or people? I think that the easier way would be to confirm the results with simple non-naturalistic images which include only walls without objects.

2. Another worry is in eye movements: Bigger nearby surfaces may attract fixations, modulating responses via attention rather than distance per se. The authors could quantify eye movements (e.g. number of fixations, average fixation length, etc.) using the DeepMReye software even if they did not collect eye movements by design.

3. It is not clear why the number of visual ROIs considered diminishes in the second experiment. A strong justification is needed. Were the ROIs pre-registered? Results are the sames (in terms of statistics and multiple comparisons) if all the ROIs are considered?

Reviewer #3: This paper provides a timely investigation about how the human visual system represents 3D spatial layout from naturalistic scenes. The approach that the authors are taking is interesting and well motivated. I think the authors are addressing an important gap in the field, which is to quantify 3D layout features from real-world naturalistic scenes, because most of the quantifications on naturalistic large-scale dataset was focused on objects, object co-occurrence and/or lower-level computer-vision based image features. In addition to using a novel computer vision-based method for recovering 3D layout structure from natural images, the authors are combining fMRI and MEG, which is an interesting and valuable approach.

While the paper seem somewhat technically focused (e.g., quantification of 3D info), I found the paper to be well referenced and well situated within important theoretical frameworks in spatial cognition, such as vector-based coding of spatial layout (e.g., Bicanski & Burgess, 2020). The use of the Natural Scenes Dataset (NSD) is a great approach and sets the paper apart from other more controlled or game-based scene approaches. Because NSD does not include ground-truth 3D layout annotations, the authors' development of a computer vision pipeline to reconstruct 3D scene structure is crucial and needed. The validation on the Matterport3D dataset, which includes known camera parameters, provides confidence in the approach. One question that could be addressed more clearly is whether the method generalizes to outdoor environments, or whether its assumptions are specific to indoor scenes. Clarifying this would help contextualize the scope and limitations of the approach.

Related to above, I also wonder if the authors make the 3D analysis codes available.

The authors' representational similarity analysis (RSA) reveals an interesting dissociation between the neural encoding of distance and orientation. After statistically controlling for low-, mid-, and high-level 2D image statistics (GIST, Portilla-Simoncelli, object2vec), early visual cortex continues to track relative distance, whereas hV4, OPA, and PPA selectively track boundary orientation. One thing that I was not clear of was how the authors controlled for the low-level information to set it apart from layout information. Did the authors use regression modeling, or some other kind of orthogonalization? This wasn't clear from the paper and needs more information.

The finding that semantic features showed partial correlations even in early visual areas is interesting and deserves a sentence or two of interpretation, as it seems to underscore the extent to which even early visual cortex is influenced by high-level structure in naturalistic inputs.

The dissociation between 3D layout features (e.g., distance, orientation) and self-pose features (pitch, roll) is interesting but seems under-described in the paper. While distance and orientation is intuitively clear, self-pose features are less intuitive and may benefit from additional clarification in the introduction. Related to this point, I wonder whether the authors view self-pose features as distinct viewer-centered/egocentric parameters while view the 3D layout features as allocentric parameters.

The switch to decoding methods for self-pose features (rather than RSA, used elsewhere in the paper) would benefit from a justification.

In my opinion, one of the most interesting parts of the paper was the demonstration that encoding of distance and orientation is modulated by task demands. The comparison between a texture discrimination task and a layout discrimination task shows that OPA exhibit enhanced correlations with layout-relevant features only when spatial judgments are required. This challenges the prevailing assumption that distance and orientation are exclusively bottom-up properties inherent to the image. Instead, the results strongly suggest top-down modulation in spatial layout processing. The authors' discussion points in this direction, but I believe the manuscript would be strengthened by developing this point further. In particular, it would be useful to elaborate on why some regions show task-dependent enhancement while others remain task-invariant, and also elaborate on mechanism of how the task-specific effects might influence these regions. Theoretically strengthening this results would help increase the significance of the findings.

---

## [Editor Report · Decision Letter 2]

4 Mar 2026

Dear Dr Li,

Thank you for your patience while we considered your revised manuscript "Vectorial boundary representation of 3D layout in human visual cortex" for publication as a Research Article at PLOS Biology. This revised version of your manuscript has been evaluated by the PLOS Biology editors and the Academic Editor.

Based on our Academic Editor's assessment of your revision, we are likely to accept this manuscript for publication. Please also make sure to address the following data and other policy-related requests.

IMPORTANT: Please ensure that you address all of the following editorial points:

**Title:

-- We would like to tweak your paper's title to emphasize the main findings. Is this alternative version acceptable to you?

"Three-dimensional scene boundary representations for wall orientation and distance are represented distinctly in the human visual cortex"

**Ethics:

-- The Ethics statement needs to be a separate, independent (and the first) subheading in the Material & Methods section. Please the information about ethical approvals and consent to this new subheading.

**Data & Code:

-- Thank you for uploading your data and scripts to the OSF repository. Please generate a DOI and include this instead of the hyperlink. Please note that PLOS policy requires that all original code and scripts used in the analyses be made available. Please also select a license for your code.

-- We did not find it intuitive to link the provided files to the data displayed in the figure panels without running your code, which makes it harder for others to re-use or re-analyze your data. We therefore ask that you provide the numerical data underlying the plots, either as a new Supporting Information file, S1 Data (S1_Data.xlsx), or in your online deposition. You can place the data from each figure or plot in a separate tab. This applies to the following figure panels:

2C

4

6ABC

7C

8C

S3ABC

S5ABCD

S6ABCD

S8

S9ABC

-- Please also include in all figure legends (including supplementary figures) a note of where the underlying data can be found (e.g., in S1 Data or in the OSF deposition).

**Supplement format:

-- As supplementary documents are not proofread and are rarely examined by readers, our preference is that, as much as possible, you integrate items currently found in the supplement into the main manuscript. Either way, please include all supplementary tables and legends in the main manuscript file, and upload all supplementary figures individually as Supporting Information files.

We expect to receive your revised manuscript within two weeks.

*Published Peer Review History*

*Press*

Sincerely,

Taylor

Taylor Hart, PhD,

Associate Editor

thart@plos.org

PLOS Biology

---

## [Editor Report · Decision Letter 3]

11 Mar 2026

Dear Dr Li,

Thank you for the submission of your revised Research Article "Three-dimensional scene boundary representations for wall orientation and distance are represented distinctly in the human visual cortex" for publication in PLOS Biology. On behalf of my colleagues and the Academic Editor, Aniruddha Das, I am pleased to say that we can in principle accept your manuscript for publication, provided you address any remaining formatting and reporting issues. These will be detailed in an email you should receive within 2-3 business days from our colleagues in the journal operations team; no action is required from you until then. Please note that we will not be able to formally accept your manuscript and schedule it for publication until you have completed any requested changes.

PRESS

Sincerely,

Taylor

Taylor Hart, PhD,

Associate Editor

PLOS Biology

thart@plos.org